# Age-dependent aggregation of ribosomal RNA-binding proteins links deterioration in chromatin stability with challenges to proteostasis

Julie Paxman[1†], Zhen Zhou[1†], Richard O'Laughlin[2], Yuting Liu[1], Yang Li[1], Wanying Tian[1], Hetian Su[1], Yanfei Jiang[1], Shayna E Holness[3], Elizabeth Stasiowski[2], Lev S Tsimring[4], Lorraine Pillus[1,5], Jeff Hasty[1,2,4], Nan Hao[1,2,4]*

[1]Department of Molecular Biology, Division of Biological Sciences, University of California, San Diego, La Jolla, United States; [2]Department of Bioengineering, University of California, San Diego, La Jolla, United States; [3]Department of Chemistry and Biochemistry, University of California, San Diego, La Jolla, United States; [4]Synthetic Biology Institute, University of California, San Diego, La Jolla, United States; [5]UCSD Moores Cancer Center, University of California San, Diego, La Jolla, United States

**\*For correspondence:**
nhao@ucsd.edu

[†]These authors contributed equally to this work

**Competing interest:** The authors declare that no competing interests exist.

**Abstract** Chromatin instability and protein homeostasis (proteostasis) stress are two well-established hallmarks of aging, which have been considered largely independent of each other. Using microfluidics and single-cell imaging approaches, we observed that, during the replicative aging of *Saccharomyces cerevisiae*, a challenge to proteostasis occurs specifically in the fraction of cells with decreased stability within the ribosomal DNA (rDNA). A screen of 170 yeast RNA-binding proteins identified ribosomal RNA (rRNA)-binding proteins as the most enriched group that aggregate upon a decrease in rDNA stability induced by inhibition of a conserved lysine deacetylase Sir2. Further, loss of rDNA stability induces age-dependent aggregation of rRNA-binding proteins through aberrant overproduction of rRNAs. These aggregates contribute to age-induced proteostasis decline and limit cellular lifespan. Our findings reveal a mechanism underlying the interconnection between chromatin instability and proteostasis stress and highlight the importance of cell-to-cell variability in aging processes.

## Editor's evaluation

Investigating the link between rDNA silencing and protein homeostasis, this study addresses an interesting and exciting question. The authors show how age-dependent loss of rDNA silencing contributes to protein aggregation. Importantly, the article furthers the understanding of distinct aging trajectories and raises important questions about how these processes might be relevant in multicellular organisms.

## Introduction

Cellular aging is a complex biological phenomenon characterized by damage accumulation, leading to loss of homeostatic cellular function and ultimately cell death (*Kirkwood, 2005*; *Ogrodnik et al., 2019*). As cellular aging has been studied, macromolecular changes have been identified as hallmarks

of aging, including mitochondrial dysfunction, genomic instability, aberrant protein expression and aggregation, among others (*López-Otín et al., 2013*). It has been generally accepted that these changes occur together during aging, contributing to cellular decline and death. However, from a single-cell perspective, what remains unclear is whether these hallmarks concur in an individual aging cell. And, if so, is there a cascade of molecular events driving the aging of single cells, in which these hallmark factors interact with and influence one another to induce aging phenotypes, functional deterioration, and ultimately cell death (*Crane and Kaeberlein, 2018*; *Kirkwood and Kowald, 1997*).

We have used replicative aging of the yeast *Saccharomyces cerevisiae* as a genetically tractable model to investigate the dynamic interactions among aging-related processes (*OLaughlin et al., 2020*). Yeast replicative aging is characterized as the finite number of cell divisions before cell death (*Mortimer and Johnston, 1959*). Previous studies have identified many conserved features in yeast aging that also accompany human aging (*Janssens and Veenhoff, 2016*). Among these, chromatin instability and mitochondrial dysfunction are considered major drivers of aging in yeast. In particular, the instability of ribosomal DNA (rDNA), driven by decreased rDNA silencing, was among the first to be identified as a causal factor of yeast aging (*Defossez et al., 1999*; *Li et al., 2017*; *Saka et al., 2013*; *Sinclair and Guarente, 1997*). Furthermore, mitochondrial dysfunction, driven by early-life decreases in vacuolar acidity and intracellular heme level, was also identified as a major contributor to yeast aging (*Hughes and Gottschling, 2012*; *Li et al., 2020*; *Veatch et al., 2009*). Adding to these observations, our recent single-cell analyses revealed that rDNA instability and mitochondrial dysfunction are mutually exclusive in individual aging cells. In an isogenic population, about half of aging cells show loss of rDNA stability and nucleolar decline, whereas the other half experience mitochondrial dysfunction (*Li et al., 2020*).

A decline in proteostasis is another well-recognized hallmark of cellular aging (*Hipp et al., 2019*; *Taylor and Dillin, 2011*) and age-related diseases, including a spectrum of neurodegenerative diseases (*Höhn et al., 2020*; *Kurtishi et al., 2019*). During aging, the cell gradually loses the ability to maintain proteostasis, resulting in damaged protein accumulation, protein aggregation, and eventually cell death. However, how aging causes challenges to proteostasis and how these challenges interplay with other age-induced processes remain largely elusive. Here, we combined microfluidic platforms with advanced imaging techniques to investigate proteostasis and protein aggregation during yeast aging at the single-cell level. We observed that a challenge to proteostasis occurred almost exclusively in the aging cells that also experienced loss of rDNA stability. Because many RNA-binding proteins are aggregation-prone and hence sensitive to proteostatic changes, we performed a systematic screen for aggregation of yeast RNA-binding proteins and identified ribosomal RNA (rRNA)-binding proteins as the most enriched group of proteins that aggregate upon decreased rDNA stability. We further found that loss of rDNA stability leads to rRNA-binding protein aggregation through excessive rRNA production and that these age-induced aggregates contribute to loss of proteostasis and accelerate aging, providing a new mechanistic connection between chromatin instability and proteostasis decline.

## Results

### A challenge to proteostasis concurs with loss of rDNA stability during cell aging

To track the challenges to proteostasis in single aging cells, we monitored Hsp104-GFP, a canonical protein stress reporter that forms aggregates (visualized as fluorescent foci) upon proteotoxic stress (*Lum et al., 2004*; *Tkach and Glover, 2004*). Consistent with previous studies (*Andersson et al., 2013*; *Erjavec et al., 2007*; *Saarikangas and Barral, 2015*), yeast cells form Hsp104-GFP foci during aging, and the frequency of such appearance increases with age. However, we noticed that Hsp104-GFP foci did not appear universally in all aging cells. Instead, only a fraction of cells showed foci formation during aging (*Figure 1A*).

We recently discovered and designated two distinct forms of aging processes in isogenic yeast cells as 'Mode 1' and 'Mode 2' (*Jin et al., 2019*; *Li et al., 2020*). Mode 1 aging is driven by loss of rDNA stability and is characterized by continuous production of elongated daughter cells at the late stages of lifespan. In contrast, Mode 2 aging retains rDNA stability and is associated with production of small round daughters (*Li et al., 2020*). We classified a population of aging cells into Mode 1 and Mode 2 (see *Figure 1*, 'Materials and methods,' and *Figure 1—figure supplement 1*) and found that

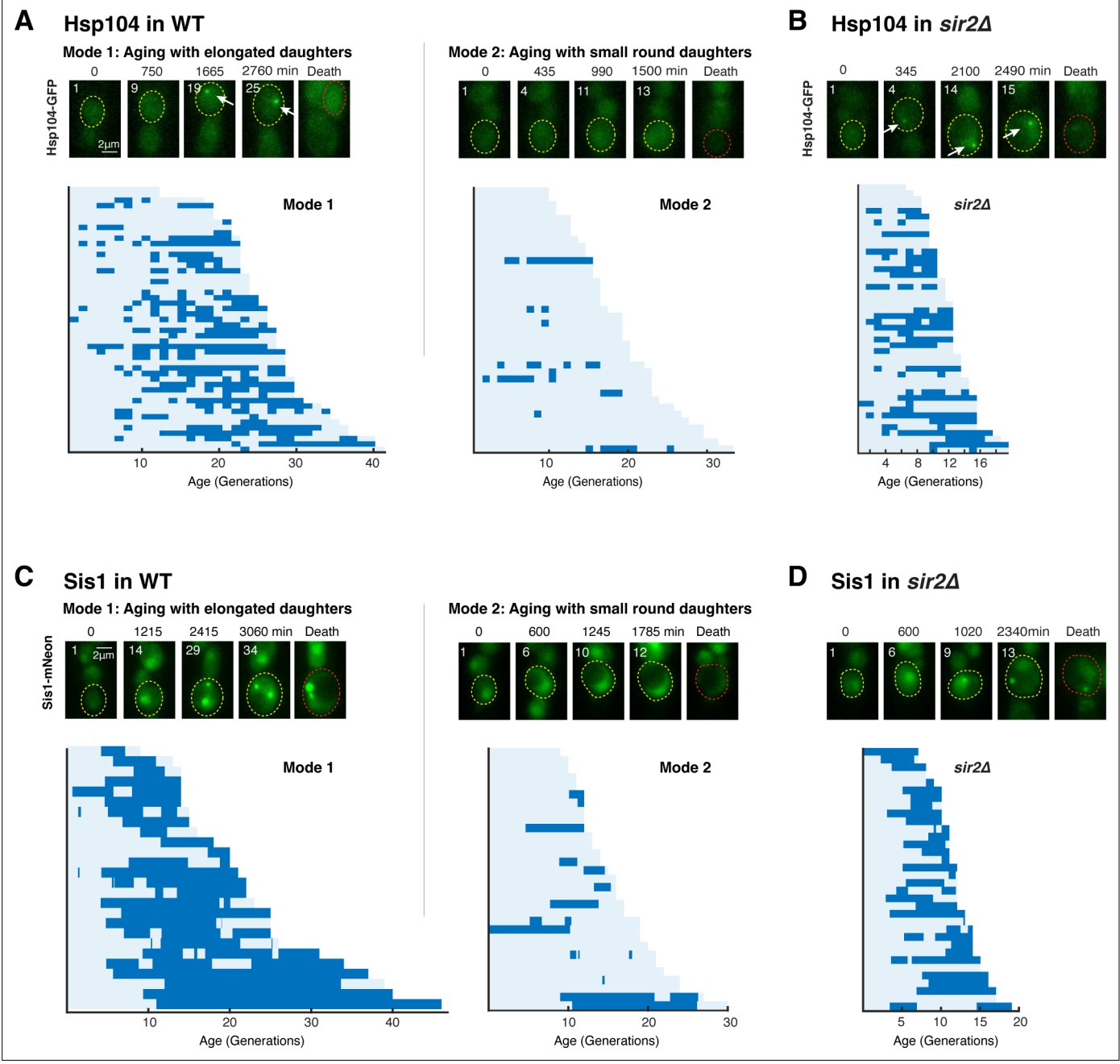

**Figure 1.** A challenge to proteostasis occurs specifically in aging cells that undergo loss of rDNA stability. (**A**) Hsp104 foci formation during aging of WT cells. Top: representative time-lapse images of Hsp104-GFP in WT Mode 1 and Mode 2 aging processes. Replicative age of mother cell is shown at the top-left corner of each image. For phase images, aging and dead mothers are marked by yellow and red arrows, respectively. In fluorescence images, aging and dead mother cells are circled in yellow and red, respectively. White arrows point to fluorescence foci of Hsp104-GFP. Mode 1 and Mode 2 cells were classified based on their age-dependent changes in their daughter morphologies (*Li et al., 2020*). Bottom: single-cell color map trajectories of Hsp104-GFP foci formation in WT Mode 1 and Mode 2 cells. Each row represents the time trace of a single cell throughout its lifespan. Color represents the absence (light blue) or presence (dark blue) of foci within a given cell cycle. Cells are sorted based on their lifespans. Single-cell color map trajectories of iRFP fluorescence and cell cycle length from the same cells are shown in *Figure 1—figure supplement 1* to confirm the classification of Mode 1 and Mode 2. (**B**) Hsp104 foci formation during aging of *sir2Δ* cells. Top: representative time-lapse images of Hsp104-GFP in *sir2Δ* cells during aging. Bottom: single-cell color map trajectories of Hsp104-GFP foci formation in *sir2Δ* cells. (**C**) Sis1 foci formation during aging of WT cells. Top: representative time-lapse images of Sis1-mNeon in WT Mode 1 and Mode 2 aging processes. Note that the expression level of Sis1 in young cells is relatively uniform and does not correlate with the cell's future aging path, Mode 1 vs. Mode 2. Bottom: single-cell color map trajectories of Sis1-mNeon foci formation in WT Mode 1 and Mode 2 cells. (**D**) Sis1 foci formation during aging of *sir2Δ* cells. Top: representative time-lapse images of Sis1-mNeon in *sir2Δ* cells during aging. Bottom: single-cell color map trajectories of Sis1-mNeon foci formation in *sir2Δ* cells.

*Figure 1 continued on next page*

*Figure 1 continued*

The online version of this article includes the following video and figure supplement(s) for figure 1:

**Figure supplement 1.** Age-dependent dynamics of iRFP fluorescence and cell cycle length used to confirm the classification of Mode 1 and Mode 2 cells.

**Figure supplement 2.** ΔssCPY*-GFP forms aggregates specifically in Mode 1 aging cells.

**Figure supplement 3.** Relationship between the age at first Hsp104 foci appearance and final lifespan in single cells.

**Figure supplement 4.** Hsp104 foci formation in *hap4Δ* and *hap4Δ sir2Δ*.

**Figure 1—video 1.** Tracking Hsp104-GFP during aging of a representative WT Mode 1 cells.

https://elifesciences.org/articles/75978/figures#fig1video1

**Figure 1—video 2.** Tracking Hsp104-GFP during aging of a representative WT Mode 2 cells.

https://elifesciences.org/articles/75978/figures#fig1video2

Hsp104-GFP form foci almost exclusively in Mode 1 aging cells, but rarely in Mode 2 cells (*Figure 1A*, *Figure 1—videos 1; 2*), indicating that a challenge to proteostasis occurs specifically in Mode 1 aging. We monitored another reporter of proteotoxic stress, ΔssCPY*-GFP. The ΔssCPY*-GFP reporter is an unstable carboxypeptidase-GFP fusion protein (*Eisele and Wolf, 2008*; *Medicherla et al., 2004*). Consistent with the observed pattern of Hsp104-GFP aggregation, ΔssCPY*-GFP formed aggregates specifically in Mode 1 aging cells (*Figure 1—figure supplement 2*).

Because Mode 1 aging is driven by rDNA instability and at the same time exhibits a response to proteotoxic stress, we speculated that rDNA stability might influence the state of proteostasis. To test this possibility, we deleted *SIR2*, which encodes a conserved lysine deacetylase that mediates chromatin silencing and stability at rDNA (*Gartenberg and Smith, 2016*), in the Hsp104-GFP reporter strain. We found that, compared to WT, a dramatically larger fraction of cells showed Hsp104 foci formation in the short-lived *sir2Δ* mutant with loss of rDNA stability (*Fritze et al., 1997*; *Figure 1B*). In addition, these foci were formed earlier in life and persisted for a larger portion of the lifespan in *sir2Δ* cells than those in WT cells (*Figure 1B*). We further observed a correlation between the time of first foci appearance and the final lifespan in WT and *sir2Δ* cells, suggesting that the proteostasis stress, indicated by Hsp104 foci formation, contributes to cell aging and death (*Figure 1—figure supplement 3*). Moreover, to exclude the possibility that *sir2Δ* exacerbates Hsp104 aggregation simply because it is short-lived, we monitored Hsp104 foci formation during the aging process in the *hap4Δ* strain, which is short-lived due to mitochondrial defects but has enhanced rDNA stability (*Li et al., 2020*). We observed strikingly decreased Hsp104 foci formation compared to WT (*Figure 1—figure supplement 4A*). Deletion of *SIR2* in the *hap4Δ* strain had a modest effect on the lifespan, but substantially increased Hsp104 foci formation (*Figure 1—figure supplement 4B*), confirming the role of Sir2 in protecting from proteostasis stress.

To determine whether proteostasis in the nucleus is similarly challenged during aging, we monitored mNeon-tagged Sis1, an Hsp40 co-chaperone and a reporter for nuclear proteostatic stress (*Feder et al., 2021*; *Klaips et al., 2020*). We found that Sis1-mNeon formed sharp foci predominantly in WT Mode 1 aging cells (*Figure 1C*) and exhibited earlier and more frequent foci formation in *sir2Δ* cells (*Figure 1D*), in agreement with Hsp104 aggregation during aging.

Taken together, these results suggest that rDNA instability can serve as a contributing factor for age-associated challenges to proteostasis. Sir2, by maintaining rDNA silencing, represses age-dependent protein aggregation, consistent with the role of sirtuins in alleviating protein aggregation-induced cytotoxicity and disorders (e.g., Huntington disease), in yeast and mammalian models (*Cohen et al., 2012*; *Jiang et al., 2011*; *Kobayashi et al., 2005*; *Sorolla et al., 2011*).

## A screen identifies rRNA-binding proteins that aggregate in response to a loss of Sir2 activity

RNA-binding proteins bind to RNAs and form ribonucleoprotein complexes that regulate the localization, processing, modification, translation, storage, and degradation of associated RNAs (*Buchan, 2014*; *Lee and Lykke-Andersen, 2013*; *Mitchell and Parker, 2014*; *Ramaswami et al., 2013*). A disproportionately high number of RNA-binding proteins contain low complexity, prion-like domains and hence are aggregation-prone (*Calabretta and Richard, 2015*; *Kato et al., 2012*; *Weber and Brangwynne, 2012*). Therefore, we reasoned that RNA-binding proteins may be especially sensitive

to the intracellular proteostasis environment and considered whether aging or loss of Sir2 activity will lead to RNA-binding protein aggregation as a driving factor for age-induced protein misfolding and proteotoxic stress.

To test this, we performed a screen to identify RNA-binding proteins that aggregate in response to a sustained loss of Sir2 activity, which induces rDNA silencing loss and mimics the later phases of Mode 1 aging. We used a recently developed synthetic genetic sensor for protein aggregation – the yeast transcriptional reporting of aggregating proteins (yTRAP) RNA-binding protein sensor library for the screen (*Newby et al., 2017*). The yTRAP RNA-binding protein sensor library is composed of 170 unique sensor strains, encompassing every known RNA-binding protein with an experimentally confirmed physical interaction with RNAs in yeast (*Figure 2—figure supplement 1A*). The aggregation state of an RNA-binding protein can be reflected by the fluorescence signal of its sensor strain. When RNA-binding proteins are in a soluble unaggregated state, the sensor GFP fluorescence is high; however, if the RNA-binding protein enters an aggregated state, the GFP fluorescence is reduced (*Figure 2A*).

To conditionally trigger a loss of Sir2 activity, we exposed cells to nicotinamide (NAM), a commonly used inhibitor of Sir2 (*Bitterman et al., 2002*; *Kato and Lin, 2014*; *Orlandi et al., 2017*). The NAM treatment induced elongated cell morphology, rDNA silencing loss (indicated by a constantly high rDNA-GFP signal) (*Li et al., 2017*), and increased Hsp104 aggregation (*Figure 2B and C*), recapitulating the aged phenotypes in *sir2Δ* and in the late phases of Mode 1 aging. To track the fluorescence changes of yTRAP RNA-binding protein sensor strains in response to NAM at high-throughput over time, we used a new version of our recently published large-scale microfluidic platform 'DynOMICS' (*Graham et al., 2020*) that was specifically modified to permit the analysis of libraries of fluorescent yeast strains. This device enables simultaneous quantitative measurements of 48 different fluorescent sensor strains over the course of several days. We observed that some sensor strains exhibited a dramatic decrease in fluorescence, indicating RNA-binding protein aggregation, upon the NAM treatment (*Figure 2D*, 'Responders'); in contrast, other sensor strains showed modest fluorescence changes, indicating minor changes in the aggregation state (*Figure 2D*, 'Non-Responder'). We note that because all the RNA-binding protein sensors in the library are under the same constitutive promoter, the fluorescence changes were specifically due to sensor aggregation state changes, not differential expression-level changes (*Newby et al., 2017*).

Of the 170 RNA-binding proteins tested on our screen (*Figure 2E*, left, and *Figure 2—figure supplement 1A*), we identified 43 Responders, which displayed at least a 50% decrease in the normalized fluorescence signal upon the NAM treatment (*Figure 2E*, middle, and *Figure 2—figure supplement 1B*), and 15 Responders with a more than 75% decrease in the normalized fluorescence signal (*Figure 2E*, right, and *Figure 2—figure supplement 1C*). From both Responder groups, we found a consistent and striking enrichment of rRNA-binding proteins involved in rRNA processing. These results indicate that rRNA-binding proteins are a major class of proteins that aggregate upon a loss of Sir2 activity.

We chose the top five rRNA-binding protein responders (Nop15, Sof1, Rlp7, Nop13, Mrd1; *Figure 2—figure supplement 1C*) for further study. To examine whether the aggregation of these proteins upon NAM treatments is mediated specifically through Sir2, we deleted the endogenous copy of *SIR2* and introduced a doxycycline-controlled promoter system for Sir2 expression in each of the yTRAP sensor strains. We showed that the absence of Sir2 expression promoted aggregation of all five rRNA-binding proteins tested (*Figure 2—figure supplement 2*), confirming the specificity of aggregation to loss of Sir2.

## Age-dependent rRNA-binding protein aggregation contributes to nuclear proteostasis stress and limits cellular lifespan

To confirm that the identified rRNA-binding proteins indeed aggregate during natural aging, we generated strains with an integrated copy of each rRNA-binding protein candidate C-terminally tagged with mNeon (*Figure 2E*, right, and *Figure 2—figure supplement 1C*). To visualize age-induced rRNA-binding protein aggregation, we tracked single aging cells using microfluidics, and then used confocal microscopy to capture high-resolution images of young cells (1 hr after loading) and aged cells (aged for 40 hr, ~80% of the average lifespan), respectively (*Figure 3A*). We observed that the rRNA-binding proteins in young cells are uniformly localized along one side of the

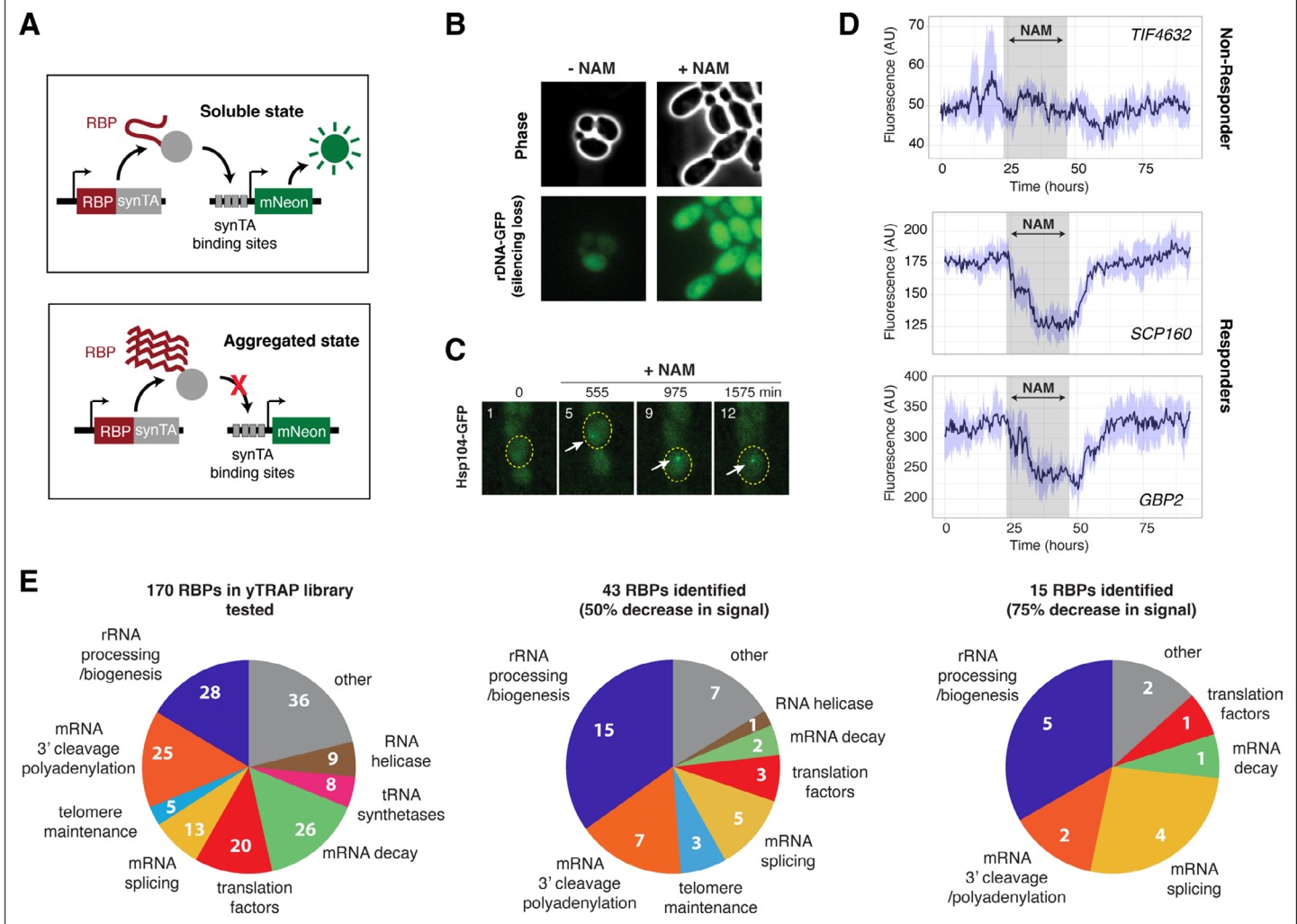

**Figure 2.** A screen identifies 27 RNA-binding proteins (RBPs) that aggregate in response to loss of Sir2 activity. (**A**) Schematic of the yTRAP synthetic genetic system that functions by coupling aggregation states of proteins to the expression of a fluorescent reporter. (**A**) has been adapted and modified from Figure 1A in *Newby et al., 2017*. (**B**) Representative images of yeast cells following 5 mM nicotinamide (NAM) treatment. Top: phase images; bottom: fluorescence images of rDNA GFP. (**C**) Representative time-lapse images of Hsp104-GFP cells treated with NAM during aging. (**D**) Representative time traces of fluorescence changes for a 'non-responder' sensor strain (top) and 'responder' sensor strains (middle, bottom). NAM induction time shown on graph in gray. Purple shades represent standard deviations of the traces. The time traces show raw fluorescence without normalization. (**E**) Functional categories of RBPs tested (left), the responders with more than 50% decrease in normalized fluorescence signal upon NAM (middle), and the responders with more than 75% decrease in normalized fluorescence signal upon NAM (right). Complete lists of RBPs tested and responder RBPs are included in *Figure 2—figure supplement 1*.

The online version of this article includes the following source data and figure supplement(s) for figure 2:

**Source data 1.** Time traces of normalized fluorescence signals for each of the 170 RBPs tested in yTRAP library.

**Figure supplement 1.** Lists of RNA-binding proteins (RBPs) in the yTRAP library tested and those identified as 'Responders' in the screen.

**Figure supplement 2.** Confirmation of dependence on Sir2 for the top five rRNA-binding protein responders from the screen.

nucleus, forming a single crescent shape, characteristic of the yeast nucleolus. At the late stages of aging, most rRNA-binding proteins formed multiple irregular-shaped coalescences or condensates (visualized as fluorescent patches or foci) in Mode 1 aged cells. In contrast, in Mode 2 aged cells, these rRNA-binding proteins all remained in the uniform crescent shape like that of young cells (*Figure 3A*). These data confirmed the yTRAP screen results (*Figure 2*) and suggested a connection between loss of rDNA silencing with age-induced rRNA-binding protein aggregation. We also observed that age-dependent condensation led to a partial loss of colocalization of rRNA-binding

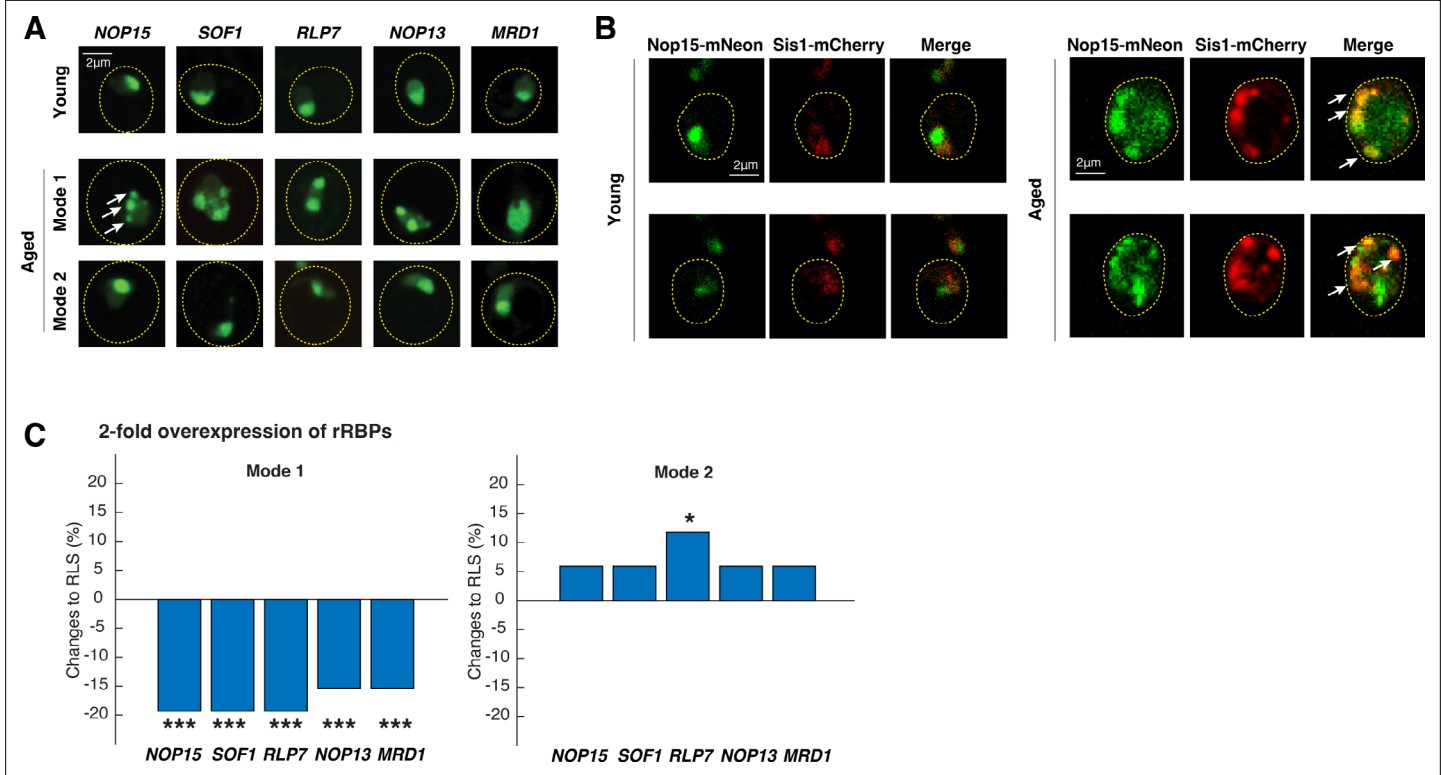

**Figure 3.** Age-dependent aggregation of rRNA-binding proteins and their effects on cellular lifespan. (**A**) Representative confocal images of rRNA-binding proteins in young cells, Mode 1 aged, and Mode 2 aged cells. Young cells and aged mother cells are circled in yellow. White arrows point to the aggregates. (**B**) Representative confocal images of Nop15-mNeon and Sis1-mCherry in young (left) and aged Mode 1 (right) cells. Young cells and aged mother cells are circled in yellow. White arrows point to the colocalized aggregates. (**C**) The effects of twofold overexpression of each rRNA-binding protein on the lifespans of Mode 1 (left) and Mode 2 (right) cells. The percentage changes of the mean replicative lifespan (RLS) relative to that of WT have been shown in the bar graphs. Asterisk indicates the significance of the changes: ***p<0.001; *p<0.05. The complete RLS curves and p-values are shown in *Figure 3—figure supplement 2*.

The online version of this article includes the following figure supplement(s) for figure 3:

**Figure supplement 1.** Partial loss of colocalization of Nop13 and Nop15 upon aggregation in Mode 1 aged cells.

**Figure supplement 2.** Replicative lifespans (RLSs) of Mode 1 and Mode 2 cells upon twofold overexpression of rRNA-binding proteins.

proteins (e.g., Nop13 and Nop15; *Figure 3—figure supplement 1*), which may be indicative of a deterioration of their coordinated functions in rRNA processing and ribosomal biogenesis in the nucleolus.

To determine whether the rRNA-binding protein condensates we observed in aged cells are indeed aggregates, we monitored the localization of Nop15, a representative rRNA-binding protein, and Sis1, the Hsp40 co-chaperone that functions in clearance of misfolded proteins in the nucleus (*Feder et al., 2021*; *Klaips et al., 2020*). We found that Sis1 clearly accumulated in Nop15 condensates in aged cells, whereas no such colocalization was observed in young cells (*Figure 3B*). These results indicate that age-induced rRNA-binding protein condensates are bona fide protein aggregates and contribute to the challenges to nuclear proteostasis observed during aging (as visualized by Sis1 foci; *Figure 1C*).

To determine the effect of rRNA-binding protein aggregation on lifespan, we overexpressed each of the rRNA-binding proteins, which leads to increased aggregation based on the law of mass action. For each of the rRNA-binding proteins tested, we observed consistently that twofold overexpression of each significantly shortened the lifespan of Mode 1 aging cells, but not that of Mode 2 aging cells in the same isogenic population (*Figure 3C* and *Figure 3—figure supplement 2*). These results indicate that the aggregated form (in Mode 1 cells) of these rRNA-binding proteins causes cell deterioration and limits cellular lifespan, whereas increasing the non-aggregated form (in Mode 2 cells) shows either no effect or a small extension of the lifespan.

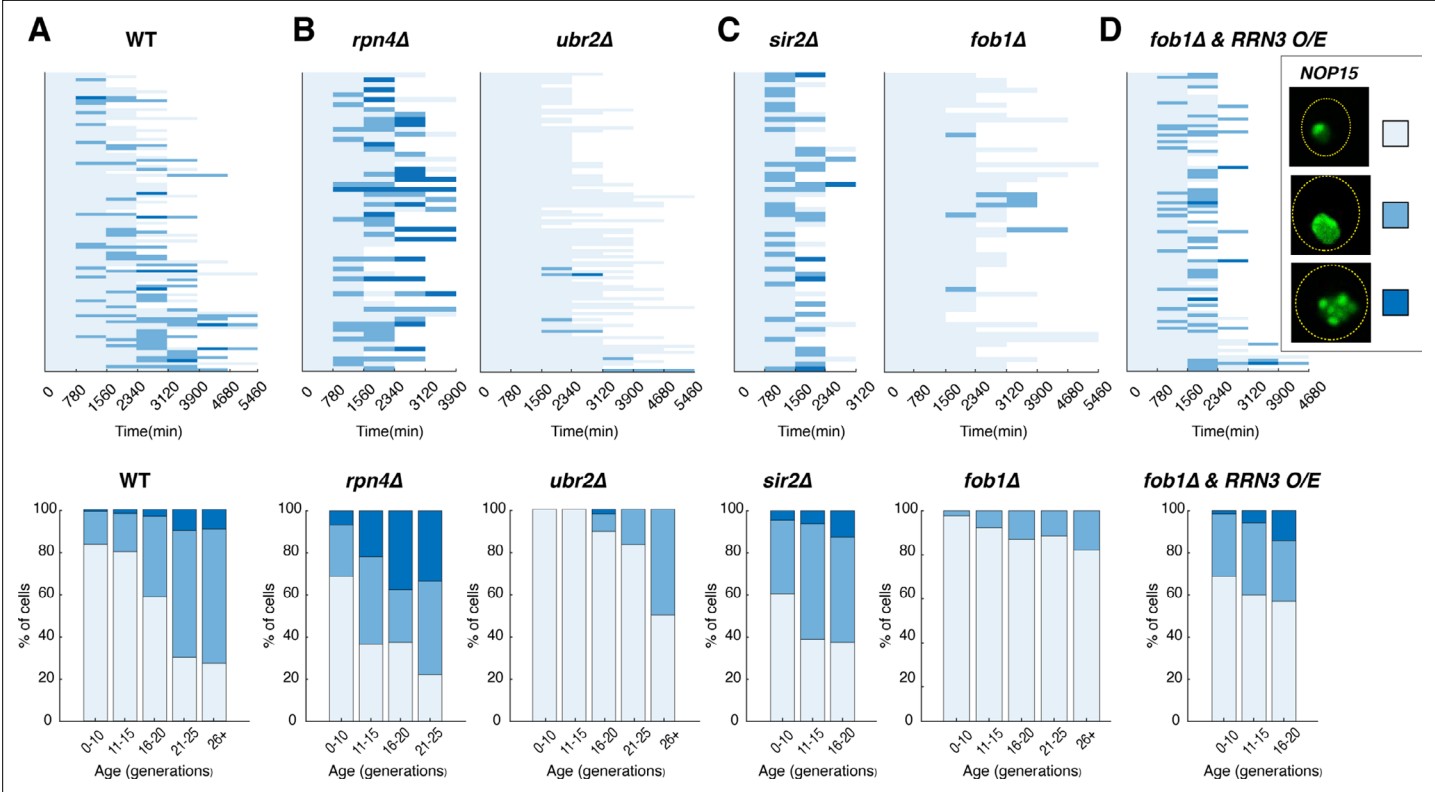

**Figure 4.** Age-dependent aggregation of Nop15 in various mutants. Single-cell color map trajectories indicate the timing and extent of age-dependent Nop15 aggregation in (**A**) WT, (**B**, left) *rpn4Δ*, (**B**, right) *ubr2Δ*, (**C**, left) *sir2Δ*, (**C**, right) *fob1Δ*, and (**D**) *fob1Δ + RRN3* overexpression (O/E). Each row tracks the aggregation state of a single aging cell. Confocal images were acquired at indicated time points during aging experiments. As indicated in the legend on the right, the aggregation state of Nop15 in each aging cell was classified as 'no aggregation' – evenly distributed fluorescence with a normal crescent shape (light blue), 'moderate aggregation' – unevenly distributed fluorescent patches with irregular shapes (blue), or 'severe aggregation' – multiple distinct fluorescent foci (dark blue). Bottom panels: bar charts show the percentage of cells in each aggregation state as a function of age, quantified from the data in corresponding top panels. For *RRN3* O/E, a TetO-inducible *RRN3* construct was integrated into the *fob1Δ* strain. Upon loading into the device, cells were exposed to 2 μM doxycycline to induce *RRN3* overexpression throughout the aging experiment. As a control, *fob1Δ* cells without the inducible *RRN3* construct were exposed to doxycycline to exclude the possibility that the drug treatment itself causes enhanced aggregation (*Figure 4—figure supplement 2*).

The online version of this article includes the following figure supplement(s) for figure 4:

**Figure supplement 1.** Nop15 aggregation tracks the age-induced changes in rDNA copy numbers.

**Figure supplement 2.** The effect of doxycycline on Nop15 aggregation.

## Excessive rRNA production induces rRNA-binding protein aggregation

We next considered the mechanism underlying rRNA-binding protein aggregation during aging and, in particular, how Sir2 and rDNA silencing, which primarily function in maintaining chromatin stability, influence the aggregation process. Since age-dependent aggregation and the effects on cellular lifespan were consistent for all the rRNA-binding proteins tested (*Figure 3*), we chose to perform in-depth genetic analysis of one. We selected Nop15, which functions in 60S ribosomal biogenesis, as a representative to investigate the pathways and factors that regulate rRNA-binding protein aggregation. To monitor age-dependent progression of aggregation in single cells, we tracked the aging processes of a large number of individual cells using microfluidics and time-lapse microscopy (phase images acquired every 15 min), and, in the same experiment, visualized Nop15-mNeon aggregation using confocal microscopy every 13 hr throughout the entire lifespans. We observed that Nop15 formed aggregates during aging of WT cells, with the frequency and severity increasing with age (*Figure 4A*) and following changes in rDNA copy number (*Figure 4—figure supplement 1*).

Recent studies have shown that RNA-binding proteins, many of which contain intrinsically disordered domains, are frequent substrates of proteasomal degradation (*Myers et al., 2018*; *Thapa*

*et al., 2020*). The proteasome, in cooperation with disaggregases, functions to remove misfolded or aggregated proteins (*Pohl and Dikic, 2019*). To examine the effects on rRNA-binding protein aggregation, we began with Rpn4, a transcriptional regulator of the 26S proteasome components that is required for normal levels of proteasome activity (*Xie and Varshavsky, 2001*). In cells lacking Rpn4, which are characterized by a reduced proteasome pool (*Ju et al., 2004*; *Xie and Varshavsky, 2001*), we found increased, earlier, and more severe Nop15 aggregation during aging (*Figure 4B*, left). In contrast, deletion of *UBR2*, which encodes a ubiquitin ligase that mediates Rpn4 degradation, leads to elevated proteasome capacity (*Wang et al., 2004*) and thereby dramatically alleviated Nop15 aggregation (*Figure 4B*, right). These results suggest that the proteasome participates in the process of removing age-induced rRNA-binding protein aggregates, probably through degradation of intrinsically disordered protein monomers as in the case of other misfolded protein aggregates or aberrant RNA-binding protein aggregates (*Berke and Paulson, 2003*; *Brown and Kaganovich, 2016*; *Hjerpe et al., 2016*; *Reiss et al., 2020*; *Thapa et al., 2020*).

We next examined the roles of Sir2 and rDNA stability on rRNA-binding protein aggregation. As shown in *Figure 3A*, rRNA-binding protein aggregates formed in Mode 1 aged cells, characterized with loss of rDNA silencing. Consistently, we observed much earlier and more frequent appearance of Nop15 aggregation during aging of *sir2Δ* cells (*Figure 4C*, left), indicating that deletion of Sir2 or loss of rDNA stability promotes rRNA-binding protein aggregation.

The yeast rDNA contains 100–200 tandemly arrayed copies of a 9.1 kb rDNA repeat, coding for rRNA subunits of ribosomes. Whereas rDNA is the site of active RNA polymerase (Pol) I-mediated rRNA transcription, it is also one of the three heterochromatin regions in yeast that are subject to distinct forms of transcriptional silencing (*Smith and Boeke, 1997*). During aging, loss of silencing at the rDNA enhances the rate of DNA double-strand breaks and recombination (*Lindstrom et al., 2011*; *Saka et al., 2013*), leading to the formation of extrachromosomal rDNA circles (ERCs) excised from this fragile genomic site. ERCs are self-replicating DNA circles, asymmetrically segregated to mother cells during cell division. As a result, ERCs accumulate exponentially in aging mother cells and have been proposed to be a causal factor of aging (*Sinclair and Guarente, 1997*). To determine whether loss of Sir2 or loss of rDNA stability drives rRNA-binding protein aggregation through ERC accumulation, we monitored Nop15 aggregation in the absence of Fob1, a replication fork-barrier protein that, when deleted, prevents rDNA recombination and abolishes ERC formation (*Defossez et al., 1999*; *Johzuka and Horiuchi, 2002*). We observed a dramatic reduction in Nop15 aggregation in the *fob1Δ* strain, indicating that ERC accumulation can be a major driver of age-dependent rRNA-binding protein aggregation (*Figure 4C*, right).

ERCs can impact various aspects of cellular functions, such as cell cycle progression (*Neurohr et al., 2018*) and nuclear pore integrity (*Denoth-Lippuner et al., 2014*). However, how ERCs mechanistically limit cellular lifespan is not clear. A recent study showed that ERC accumulation dramatically increases the number of transcriptionally active rDNA copies, resulting in a massive increase in pre-rRNA levels in the nucleolus. These pre-rRNAs, however, cannot mature into functional ribosomes (*Morlot et al., 2019*). Because increasing the level of RNA content generally promotes phase transition and aggregation of ribonucleoprotein complexes (*Lin et al., 2015*; *Zhang et al., 2015*), we hypothesized that excessive production of rRNAs could induce age-dependent aggregation of rRNA-binding proteins. To test this, in the *fob1Δ* mutant where ERC formation is abolished, we overexpressed Rrn3, the RNA Pol I-specific transcription factor that promotes rRNA transcription (*Moorefield et al., 2000*; *Philippi et al., 2010*; *Yamamoto et al., 1996*). In support of our hypothesis, we observed that the excessive rRNA production by Rrn3 overexpression is sufficient to induce Nop15 aggregation in the absence of ERCs (*Figure 4D* and *Figure 4—figure supplement 2*).

Taken together, these results revealed that age-dependent loss of rDNA stability promotes rRNA-binding protein aggregation through ERC accumulation and, more specifically, excessively high levels of rRNAs transcribed from ERCs. This aggregation may impair the normal function of rRNA-binding proteins in pre-rRNA processing and maturation and the assembly of functional ribosomes, accounting for the decoupling of rRNA transcription and ribosomal biogenesis during aging (*Morlot et al., 2019*).

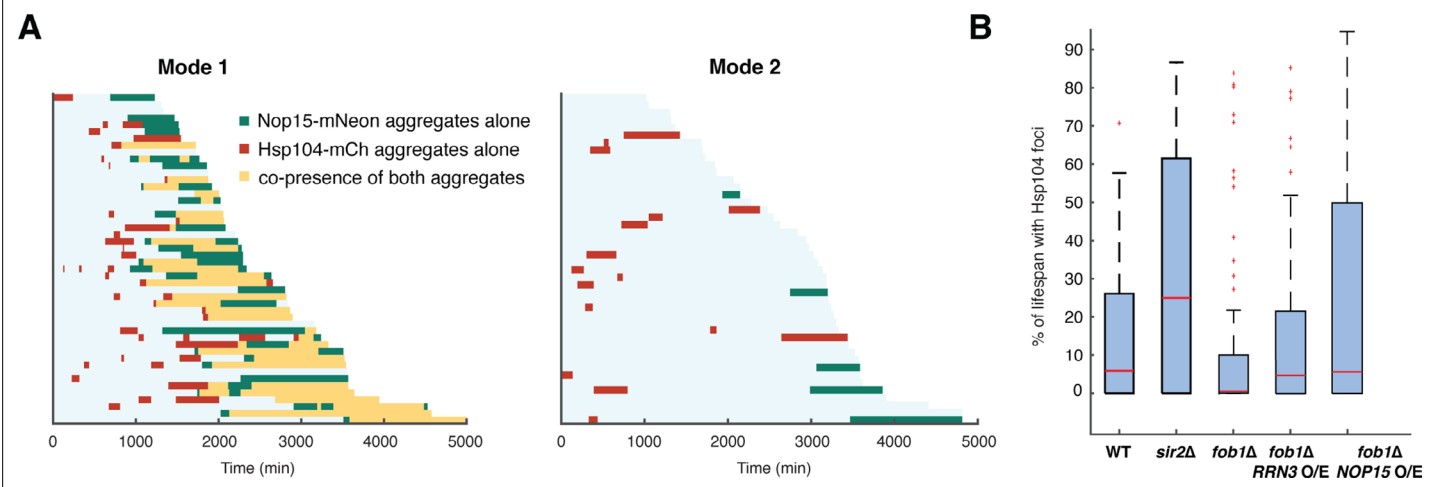

**Figure 5.** Elevated rRNA-binding protein aggregation promotes Hsp104-GFP foci formation during aging. (**A**) Single-cell color map trajectories of Nop15-mNeon aggregation (green), Hsp104-mCherry foci (red), and co-presence of both aggregates (yellow) in WT Mode 1 and Mode 2 cells. Each row represents the time trace of a single cell throughout its lifespan. Cells are sorted based on their lifespans. (**B**) Boxplots show the distributions of percentage of lifespan with Hsp104-GFP foci appearance in single aging cells for WT (n = 87), *sir2Δ* (n = 46), *fob1Δ* (n = 60), *fob1Δ + RRN3* overexpression (n = 117), and *fob1Δ + NOP15* overexpression (n = 130). In the plot, the bottom and top of the box are first (the 25th percentile of the data, q1) and third quartiles (the 75th percentile of the data, q3); the red band inside the box is the median; the whiskers cover the range between q1-1.5x(q3-q1) and q3 + 1.5x (q3–q1). The *RRN3* O/E and NOP15 O/E experiments were conducted as in *Figure 4D*.

The online version of this article includes the following video and figure supplement(s) for figure 5:

**Figure supplement 1.** Single-cell color map trajectories for Hsp104 foci formation in (**A**) *fob1Δ*, (**B**) *fob1Δ + RRN3* o/e, and (**C**) *fob1Δ + NOP15* ole.

**Figure 5—video 1.** Visualizing Hsp104-mCherry and Nop15-mNeon during aging of a representative Mode 1 cell.
https://elifesciences.org/articles/75978/figures#fig5video1

**Figure 5—video 2.** Visualizing Hsp104-mCherry and Nop15-mNeon during aging of a representative Mode 2 cell.
https://elifesciences.org/articles/75978/figures#fig5video2

## Elevated rRNA-binding protein aggregation contributes to global proteostasis stress

Previous studies showed that increased expression and aggregation of intrinsically disordered proteins impose an increased burden on protein folding resources and the proteasome, leading to global proteostasis decline (*Andersson et al., 2013*; *Bence et al., 2001*; *Outeiro and Lindquist, 2003*; *Stefani and Dobson, 2003*; *Verhoef, 2002*; *Yu et al., 2019*). To determine the relationship between rRNA-binding protein aggregation and global proteostasis stress during aging, we monitored Nop15-mNeon and Hsp104-mCherry in the same cells (*Figure 5A*). We found continuous co-occurrence of Nop15 aggregation and Hsp104 foci during the later stage of aging in the majority of Mode 1 cells (*Figure 5A*, left). Furthermore, in 66% of these cells, Nop15 aggregation (indicated by green bars in *Figure 5A*) immediately preceded the co-occurrence phase (indicated by yellow bars in *Figure 5A*), suggesting that rRNA-binding protein aggregation leads to global proteostasis stress in a large fraction of aging cells. We also observed occasional, transient appearance of Hsp104 foci during the early stage of aging in both Mode 1 and Mode 2 cells, which did not show any obvious relationship to Nop15 aggregation that occurred much later in aging (see 'Discussion').

In further support of the causal connection from rRNA-binding protein aggregation to proteostasis stress, the deletion of *SIR2*, which reduces rDNA stability and enhances rRNA-binding protein aggregation (*Figure 4C*, left), promotes proteostasis stress during aging, as reflected by increased Hsp104 foci formation (*Figures 1B and 5B*). In contrast, the deletion of *FOB1*, which enhances rDNA stability and reduces rRNA-binding protein aggregation (*Figure 4C*, right), showed dramatically decreased Hsp104 foci formation (*Figure 5B*). To test whether elevated rRNA-binding protein aggregation can impact proteostasis independent of ERCs, in the *fob1Δ* mutant where ERC formation is abolished, we overexpressed Rrn3 and Nop15, respectively, as both perturbations can enhance rRNA-binding protein aggregation. We observed increased Hsp104 foci formation, indicating that

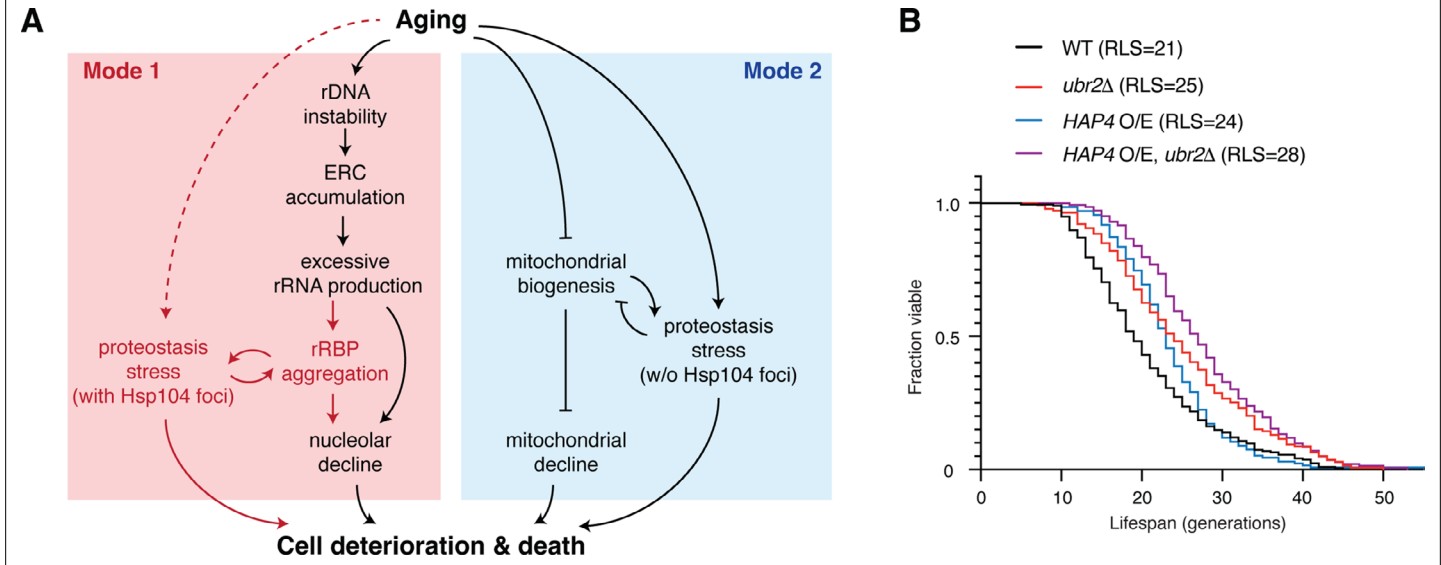

**Figure 6.** Challenges to proteostasis interact with the rDNA instability pathway and contribute to Mode 1 aging. (**A**) A schematic depicts a working model for the divergent pathways underlying single-cell aging in yeast. Red portions highlight newly identified processes and interactions in this study. (**B**) Replicative lifespans (RLSs) for WT (n = 216), *ubr2Δ* (n = 139), *HAP4* O/E (n = 134), and *HAP4* O/E, *ubr2Δ* (n = 143). RLSs shown indicate the mean lifespans.

elevated rRNA-binding protein aggregation is sufficient to induce proteostasis stress (*Figure 5B* and *Figure 5—figure supplement 1*, compare *fob1Δ + RRN3* o/e and *fob1Δ + NOP15* o/e with *fob1Δ* alone).

Taken together, our results revealed a sequential cascade of interconnected molecular events that underlies the aging process in a fraction (Mode 1) of yeast cells (*Figure 6A*): age-dependent loss of rDNA stability results in ERC accumulation (*Sinclair and Guarente, 1997*) and consequently excessive rRNA production (*Morlot et al., 2019*), which promotes aggregation of rRNA-binding proteins (*Figure 4*). These aggregates contribute to age-associated challenges to proteostasis (*Figure 5*), probably by exacerbating ribosomal dysfunction (*Henras et al., 2015*; *Woolford and Baserga, 2013*) and increasing the proteostasis burden (*Figures 3B and 4B*). The proteostasis network, in turn, regulates rRNA-binding protein aggregation via chaperone- (e.g., Sis1) (*Figures 1C and D and 3B*) and proteasome-mediated removal of protein aggregates (*Figure 4B*). The other fraction (Mode 2) of isogenic cells undergo heme depletion and mitochondrial decline during aging (*Li et al., 2020*), but not rRNA-binding protein aggregation (*Figure 3*). Overexpression of *HAP4* enhances mitochondrial biogenesis and drives the majority of cells to Mode 1 aging (*Li et al., 2020*). We found that deletion of *UBR2*, which increases proteasome capacity and alleviates rRNA-binding protein aggregation and proteostasis burden (*Figure 3B*), substantially extended the lifespan of the *HAP4* O/E strain (*Figure 6B*), in support of a working model in which proteostasis stress is a major contributing factor to Mode 1 aging driven by rDNA instability.

## Discussion

Chromatin instability and proteostasis stress are two commonly described hallmarks of aging, which have been previously considered independent of each other. However, an increasing number of recent studies suggest that these two processes may be interconnected. For example, Sir2, a conserved deacetylase encoded by the best-studied longevity gene to date, mediates deacetylation and silencing of heterochromatin regions and serves as a major regulator of chromatin stability and lifespan in yeast. Deletion of Sir2, which causes a loss of chromatin stability, dramatically elevates damaged protein accumulation and aggregation (*Aguilaniu et al., 2003*; *Cohen et al., 2012*; *Erjavec et al., 2007*), suggesting potential interplays between processes that mediate chromatin stability and proteostasis. In line with this, similar patterns of co-regulation have been observed in mammals upon perturbations that target the sirtuin family of deacetylases (*Min et al., 2010*; *Park et al., 2020*; *Tomita*

*et al., 2015*; *Westerheide et al., 2009*). However, the molecular basis underlying these connections has remained largely unclear.

In this study, we exploited the power of single-cell imaging technologies, which enabled us to track the state of proteostasis throughout lifespans of a large number of single yeast cells. Interestingly, we observed that a challenge to proteostasis, visualized by Hsp104 and Sis1 aggregation, occurs specifically in the fraction of aging cells that undergo loss of rDNA stability (Mode 1), and the *sir2Δ* mutant with decreased rDNA stability show accelerated and exacerbated protein aggregation. We further found that loss of rDNA stability causes age-dependent aggregation of rRNA-binding proteins through aberrant overproduction of rRNAs. These aggregates impair nucleolar integrity and promote proteostasis decline in aged cells. We noted that some cells showed transient Hsp104 foci very early in their lifespans (*Figures 1A and 5A*). We speculate that these aggregates might be caused by spontaneous early-life molecular or cellular changes, such as vacuolar pH changes (*Hughes and Gottschling, 2012*) or oxidative stress (*Hanzén et al., 2016*). Aggregation of rRNA-binding proteins occurs during the later stages of aging, contributing to the increasing frequency of Hsp104 aggregate appearance with age and limiting the cellular lifespan. Importantly, throughout our analyses, the fraction of aging cells that undergo mitochondrial dysfunction (Mode 2) did not show rRNA-binding protein aggregation and therefore provided a powerful isogenic control for evaluating the regulation and effects of age-induced aggregation.

Aging and aging-related processes can also be associated with other types of proteostasis stress that do not trigger Hsp104 aggregation. In fact, we observed that *ubr2Δ*, which increases proteasome capacity, extended the lifespan of Mode 1 (Mode 1 replicative lifespans [RLSs] in *ubr2Δ* vs. WT: 29 vs. 26) as well as that of Mode 2 cells that undergo mitochondrial biogenesis decline (Mode 2 RLSs in *ubr2Δ* vs. WT: 20 vs. 17), suggesting potential interactions between proteostasis stress and mitochondrial biogenesis (*Figure 6A*). Indeed, a specific mitochondrial chaperone system, which mediates import, folding, and removal of aggregation-prone mitochondrial proteins, plays an important role in maintaining proteostasis in mitochondria during aging (*Baker et al., 2011*; *Liu et al., 2022*; *Moehle et al., 2019*).

Our findings establish a mechanistic connection between chromatin stability and proteostasis stress during aging and highlight the importance of cell-to-cell variability when considering aging hallmarks and longevity-modulating perturbations. In addition, our working model can help interpret some previous intriguing yet unresolved observations. For instance, it has been puzzling why overexpression of Hsp104, a disaggregase that clears protein aggregation, can dramatically extend the lifespan of *sir2Δ*, but not WT cells (*Erjavec et al., 2007*). Based on our results, only a fraction of WT cells (Mode 1) experience chromatin instability and proteostasis decline at late stages of aging and therefore the effect of Hsp104 overexpression on lifespan is rather modest. In contrast, the majority of *sir2Δ* cells age with accelerated and severe rRNA-binding protein aggregation and proteostasis decline due to sustained loss of chromatin stability. As a result, Hsp104 overexpression, which alleviates protein aggregation, exhibits a much more dramatic pro-longevity effect.

It is important to acknowledge that chromatin stability could be linked with proteostasis through other mechanisms, beyond that reported in this study. For example, in *C. elegans*, age-dependent changes in epigenetic landscape and chromatin accessibility lead to a repression of the heat-shock response (HSR) that combats protein misfolding, resulting in proteostasis decline (*Labbadia and Morimoto, 2015*). During human cell senescence, the losses of silencing and repression on transposon and satellite repeat activation cause disorganized nuclear distribution of HSF1 (the central transcription factor that mediates HSR), contributing to HSR suppression and proteostasis decline (*Gaglia et al., 2020*; *Jolly et al., 2002*; *Swanson et al., 2013*; *Van Meter et al., 2014*). A very recent report showed that ERC-like DNA circles in yeast cells impair nuclear pore integrity, causing aberrant pre-mRNA nuclear export and translation and consequently contributing to loss of proteostasis (*Meinema et al., 2021*). We speculate that multiple different mechanisms may operate in parallel to mediate the interplay between chromatin stability and proteostasis during aging. Further investigation will be needed to determine the relative contributions and dynamic coordination among these mechanisms.

We have focused our investigation of protein aggregation specifically on RNA-binding proteins. Recently, substantial interest has focused on the emerging connections between RNA-binding protein aggregation and age-related diseases. For example, aberrant formation and persistence of RNA-binding protein aggregates have been found to contribute to degenerative diseases, including

multi-system proteinopathy, Paget's disease, amyotrophic lateral sclerosis, frontotemporal lobar degeneration, and Alzheimer's disease, making RNA-binding protein aggregates promising therapeutic targets for treating degenerative diseases (*Khalil et al., 2018*; *Li et al., 2013*; *Ramaswami et al., 2013*). However, a systematic analysis of age-dependent RNA-binding protein aggregation remained missing, in part due to a lack of quantitative and high-throughput cellular reporters. In this study, we took advantage of a recently developed synthetic genetic system, yTRAP, which couples protein aggregation states to fluorescent reporter output (*Newby et al., 2017*), and a high-throughput microfluidic platform, DynOMICS, which enables simultaneous time-lapse measurements of a large number of fluorescent strains over an extended period of time (*Graham et al., 2020*). We screened the yTRAP RNA-binding protein sensor library, which encompasses most of the confirmed yeast RNA-binding proteins, and identified rRNA-binding proteins as the most enriched group of proteins that aggregate in response to loss of Sir2 activity. These rRNA-binding protein candidates have been independently confirmed and further analyzed, uncovering a mechanistic link between chromatin instability and proteostasis decline.

In addition to the rRNA-binding proteins, our screen has also identified a number of mRNA-binding proteins (*Figure 2E* and *Figure 2—figure supplement 1*). These proteins bind to mRNAs and form ribonucleoprotein complexes, which play important roles in post-transcriptional control of gene expression and a wide array of physiological functions (*Buchan, 2014*; *Chakravarty et al., 2020*; *Jiang et al., 2020*; *Lee and Lykke-Andersen, 2013*; *Mitchell and Parker, 2014*; *Ramaswami et al., 2013*). Age-dependent aggregation of mRNA-binding proteins may affect their regulatory functions, contributing to aging phenotypes and cellular decline. For example, previous studies showed that the yeast RBP Whi3 forms aggregates during aging, resulting in sterility in aged yeast cells (*Schlissel et al., 2017*). Future studies will be poised to confirm age-induced aggregation of each mRNA-binding protein identified from our screen and determine the regulation and consequence of their aggregation. In particular, proteins involved in pre-mRNA splicing are highly enriched (*Figure 2E*) and would be of special interest for in-depth analyses. These investigations, combined with our current analyses of rRNA-binding proteins, will lead to a comprehensive understanding of the role of RNA-binding protein aggregation in yeast aging.

Research on aging biology has benefited tremendously from the development of genomic sequencing and systematic analyses, which have revealed many conserved genes and hallmarks that change during cell aging and influence cellular lifespan (*Hendrickson et al., 2018*; *Janssens et al., 2015*; *López-Otín et al., 2013*; *McCormick et al., 2015*; *McCormick and Promislow, 2018*). An emerging challenge is to understand how these genes and factors interact and operate collectively to drive the aging process. Furthermore, it has been increasingly recognized that cell aging is a highly dynamic and stochastic process in which isogenic cells age with distinct molecular and phenotypic changes, and thereby markedly different lifespans. Such complexity has hindered progress in the mechanistic understanding of aging biology and the rational design of effective intervention strategies to promote longevity. Recent advances in single-cell technologies provide enabling tools for tracking the aging processes of a large number of individual cells and defining the core aging networks that govern the fate and dynamics of deterioration in aging cells (*He et al., 2020*; *OLaughlin et al., 2020*). As an example, the study presented here showcases how these technologies were applied to identify a cascade of events linking chromatin instability with proteostasis decline, two major age-induced processes in single cells, which allowed us to propose a simple molecular network that underlies single-cell aging in yeast populations. Importantly, this network model can be used to guide the design of combinatorial perturbations that target multiple core network nodes, rather than single genes, to dramatically extend cellular lifespan. We envision that systems-level single-cell analyses will become increasingly appreciated and adopted in aging research to unravel comprehensive regulatory networks that determine aging dynamics and advance a mechanistic understanding of the causes, progression, and consequences of aging.

## Materials and methods

**Key resources table**

| Reagent type (species) or resource | Designation | Source or reference | Identifiers | Additional information |
|---|---|---|---|---|
| Recombinant DNA reagent | pRS306-$P_{NOP15}$-NOP15-mNeon-$T_{ADH1}$ | This study | NHB0904 | See 'Strain and plasmid construction' for details |
| Recombinant DNA reagent | pRS306-$P_{NOP15}$-NOP15-$T_{ADH1}$ | This study | NHB0902 | See 'Strain and plasmid construction' for details |
| Recombinant DNA reagent | pRS306-$P_{SOF1}$-SOF1-mNeon-$T_{ADH1}$ | This study | NHB0892 | See 'Strain and plasmid construction' for details |
| Recombinant DNA reagent | pRS306-$P_{SOF1}$-SOF1-$T_{ADH1}$ | This study | NHB0901 | See 'Strain and plasmid construction' for details |
| Recombinant DNA reagent | pRS306-$P_{RLP7}$-RLP7-$T_{ADH1}$ | This study | NHB0896 | See 'Strain and plasmid construction' for details |
| Recombinant DNA reagent | pRS306-$P_{NOP13}$-NOP13-mNeon-$T_{ADH1}$ | This study | NHB0893 | See 'Strain and plasmid construction' for details |
| Recombinant DNA reagent | pRS306-$P_{NOP13}$-NOP13-$T_{ADH1}$ | This study | NHB0903 | See 'Strain and plasmid construction' for details |
| Recombinant DNA reagent | pRS306-$P_{MRD1}$-MRD1-mNeon-$T_{ADH1}$ | This study | NHB0927 | See 'Strain and plasmid construction' for details |
| Recombinant DNA reagent | pRS306-$P_{MRD1}$-MRD1-mNeon-$T_{ADH1}$ | This study | NHB0895 | See 'Strain and plasmid construction' for details |
| Recombinant DNA reagent | $P_{RPL18B}$-rtTA3-$T_{ADH1}$-TetO7-$P_{LEU2m}$-RRN3-mRuby2-$T_{ENO2}$-LEU2 | This study | NHB1148 | See 'Strain and plasmid construction' for details |
| Recombinant DNA reagent | $P_{RPL18B}$-rtTA3-$T_{ADH1}$-TetO7-$P_{LEU2m}$-NOP15-$T_{ENO2}$-LEU2 | This study | NHB1150 | See 'Strain and plasmid construction' for details |
| Strain, strain background (*Saccharomyces cerevisiae*) | *BY4741 MATa his3Δ1 leu2Δ0 met15Δ0 ura3Δ0, NHP6a-iRFP-kanMX, HSP104-GFP-HIS3* | This study | NHGFP0068 | See 'Strain and plasmid construction' for details |
| Strain, strain background (*S. cerevisiae*) | *BY4741 MATa his3Δ1 leu2Δ0 met15Δ0 ura3Δ0, NHP6a-iRFP-kanMX, HSP104-GFP-HIS3, sir2::CgURA3* | This study | NH0761 | See 'Strain and plasmid construction' for details |
| Strain, strain background (*S. cerevisiae*) | *BY4741 MATa his3Δ1 leu2Δ0 met15Δ0 ura3Δ0, NHP6a-iRFP-kanMX, HSP104-GFP-HIS3, fob1::CgURA3* | This study | NH0778 | See 'Strain and plasmid construction' for details |
| Strain, strain background (*S. cerevisiae*) | *BY4741 MATa his3Δ1 leu2Δ0 met15Δ0 ura3Δ0, NHP6a-iRFP-kanMX, SIS1-mNeon-URA3* | This study | NH1324 | See 'Strain and plasmid construction' for details |
| Strain, strain background (*S. cerevisiae*) | *BY4741 MATa his3Δ1 leu2-1 met15Δ0 ura3-1, NHP6a-iRFP-kanMX, ura3-1::$P_{PRC1}$-ΔssCPY*-GFP-URA3* | This study | NH1036 | See 'Strain and plasmid construction' for details |
| Strain, strain background (*S. cerevisiae*) | *BY4741 MATa his3Δ1 leu2Δ0 met15Δ0 ura3Δ0, NHP6a-iRFP-kanMX, NOP15::$P_{NOP15}$-NOP15-mNeon-$T_{ADH1}$-URA3* | This study | NH1212 | See 'Strain and plasmid construction' for details |
| Strain, strain background (*S. cerevisiae*) | *BY4741 MATa his3Δ1 leu2Δ0 met15Δ0 ura3Δ0, NHP6a-iRFP-kanMX, SOF1::$P_{SOF1}$-SOF1-mNeon-$T_{ADH1}$-URA3* | This study | NH1213 | See 'Strain and plasmid construction' for details |
| Strain, strain background (*S. cerevisiae*) | *BY4741 MATa his3Δ1 leu2Δ0 met15Δ0 ura3Δ0, NHP6a-iRFP-kanMX, RLP7::$P_{RLP7}$-RLP7-mNeon-$T_{ADH1}$-URA3* | This study | NH1187 | See 'Strain and plasmid construction' for details |
| Strain, strain background (*S. cerevisiae*) | *BY4741 MATa his3Δ1 leu2Δ0 met15Δ0 ura3Δ0, NHP6a-iRFP-kanMX, NOP13::$P_{NOP13}$-NOP13-mNeon-$T_{ADH1}$-URA3* | This study | NH1186 | See 'Strain and plasmid construction' for details |
| Strain, strain background (*S. cerevisiae*) | *BY4741 MATa his3Δ1 leu2Δ0 met15Δ0 ura3Δ0, NHP6a-iRFP-kanMX, MRD1::$P_{MRD1}$-MRD1-mNeon-$T_{ADH1}$-URA3* | This study | NH1251 | See 'Strain and plasmid construction' for details |

| Reagent type (species) or resource | Designation | Source or reference | Identifiers | Additional information |
|---|---|---|---|---|
| Strain, strain background (*S. cerevisiae*) | BY4741 MATa his3Δ1 leu2Δ0 met15Δ0 ura3Δ0, NHP6a-iRFP-kanMX, NOP15::$P_{NOP15}$-NOP15-$T_{ADH1}$-URA3 | This study | NH1218 | See 'Strain and plasmid construction' for details |
| Strain, strain background (*S. cerevisiae*) | BY4741 MATa his3Δ1 leu2Δ0 met15Δ0 ura3Δ0, NHP6a-iRFP-kanMX, SOF1::$P_{SOF1}$-SOF1-$T_{ADH1}$-URA3 | This study | NH1216 | See 'Strain and plasmid construction' for details |
| Strain, strain background (*S. cerevisiae*) | BY4741 MATa his3Δ1 leu2Δ0 met15Δ0 ura3Δ0, NHP6a-iRFP-kanMX, RLP7::$P_{RLP7}$-RLP7-$T_{ADH1}$-URA3 | This study | NH1211 | See 'Strain and plasmid construction' for details |
| Strain, strain background (*S. cerevisiae*) | BY4741 MATa his3Δ1 leu2Δ0 met15Δ0 ura3Δ0, NHP6a-iRFP-kanMX, NOP13::$P_{NOP13}$-NOP13-$T_{ADH1}$-URA3 | This study | NH1219 | See 'Strain and plasmid construction' for details |
| Strain, strain background (*S. cerevisiae*) | BY4741 MATa his3Δ1 leu2Δ0 met15Δ0 ura3Δ0, NHP6a-iRFP-kanMX, MRD1::$P_{MRD1}$-MRD1-$T_{ADH1}$-URA3 | This study | NH1215 | See 'Strain and plasmid construction' for details |
| Strain, strain background (*S. cerevisiae*) | BY4741 MATa his3Δ1 leu2Δ0 met15Δ0 ura3Δ0, NHP6a-iRFP-kanMX, NOP15::$P_{NOP15}$-NOP15-mNeon-$T_{ADH1}$-URA3, sir2::CgHIS3 | This study | NH1477 | See 'Strain and plasmid construction' for details |
| Strain, strain background (*S. cerevisiae*) | BY4741 MATa his3Δ1 leu2Δ0 met15Δ0 ura3Δ0, NHP6a-iRFP-kanMX, NOP15::$P_{NOP15}$-NOP15-mNeon-$T_{ADH1}$-URA3, rpn4::CgHIS3 | This study | NH1478 | See 'Strain and plasmid construction' for details |
| Strain, strain background (*S. cerevisiae*) | BY4741 MATa his3Δ1 leu2Δ0 met15Δ0 ura3Δ0, NHP6a-iRFP-kanMX, NOP15::$P_{NOP15}$-NOP15-mNeon-$T_{ADH1}$-URA3, fob1::CgHIS3 | This study | NH1479 | See 'Strain and plasmid construction' for details |
| Strain, strain background (*S. cerevisiae*) | BY4741 MATa his3Δ1 leu2Δ0 met15Δ0 ura3Δ0, NHP6a-iRFP-kanMX, NOP15::$P_{NOP15}$-NOP15-mNeon-$T_{ADH1}$-URA3, ubr2::CgHIS3 | This study | NH1630 | See 'Strain and plasmid construction' for details |
| Strain, strain background (*S. cerevisiae*) | BY4741 MATa his3Δ1 leu2Δ0 met15Δ0 ura3Δ0, NHP6a-iRFP-kanMX, NOP15::$P_{NOP15}$-NOP15-mNeon-$T_{ADH1}$-URA3, leu2Δ0::$P_{RPL18B}$-rtTA3-$T_{ADH1}$-TetO7-$P_{LEU2m}$-RRN3-mRuby2-$T_{ENO2}$-LEU2, fob1::CgHIS3 | This study | NH1507 | See 'Strain and plasmid construction' for details |
| Strain, strain background (*S. cerevisiae*) | BY4741 MATa his3Δ1 leu2Δ0 met15Δ0 ura3Δ0, NHP6a-iRFP-kanMX, RDN1::NTS1-$P_{TDH3}$-GFP-URA3, ubr2::CgHIS3 | This study | NH1408 | See 'Strain and plasmid construction' for details |
| Strain, strain background (*S. cerevisiae*) | BY4741 MATa his3Δ1 leu2Δ0 met15Δ0 ura3Δ0, NHP6a-iRFP-kanMX, RDN1::NTS1-$P_{TDH3}$-GFP-URA3, HAP4::$P_{TDH3}$-HAP4-LEU2, ubr2::CgHIS3 | This study | NH1642 | See 'Strain and plasmid construction' for details |
| Strain, strain background (*S. cerevisiae*) | BY4741 MATa his3Δ1 leu2Δ0 met15Δ0 ura3Δ0, NHP6a-iRFP-kanMX, HSP104-GFP-HIS3, fob1::CgURA3, leu2Δ0:: $P_{RPL18B}$-rtTA3-$T_{ADH1}$-TetO7-$P_{LEU2m}$-RRN3-mRuby2-$T_{ENO2}$-LEU2, | This study | NH1666 | See 'Strain and plasmid construction' for details |
| Strain, strain background (*S. cerevisiae*) | BY4741 MATa his3Δ1 leu2Δ0 met15Δ0 ura3Δ0, NHP6a-iRFP-kanMX, HSP104-GFP-HIS3, fob1::CgURA3, leu2Δ0:: $P_{RPL18B}$-rtTA3-$T_{ADH1}$-TetO7-$P_{LEU2m}$-NOP15-$T_{ENO2}$-LEU2 | This study | NH1685 | See 'Strain and plasmid construction' for details |
| Strain, strain background (*S. cerevisiae*) | BY4741 MATa his3Δ1 leu2Δ0 met15Δ0 ura3Δ0, NHP6a-iRFP-kanMX, HSP104-GFP-HIS3, hap4::CgURA3 | This study | NH0815 | See 'Strain and plasmid construction' for details |
| Strain, strain background (*S. cerevisiae*) | BY4741 MATa his3Δ1 leu2Δ0 met15Δ0 ura3Δ0, NHP6a-iRFP-kanMX, HSP104-GFP-HIS3, sir2::CgURA3, hap4::CgLEU2 | This study | NH1096 | See 'Strain and plasmid construction' for details |
| Strain, strain background (*S. cerevisiae*) | yTRAP-NOP15, $P_{RPL18B}$-rtTA3-$T_{ADH1}$-TetO7-$P_{LEU2m}$-SIR2-mCherry-$T_{ENO2}$-LEU2, sir2::CgHIS3 | This study | NH1788 | See 'Strain and plasmid construction' for details |
| Strain, strain background (*S. cerevisiae*) | yTRAP-SOF1, $P_{RPL18B}$-rtTA3-$T_{ADH1}$-TetO7-$P_{LEU2m}$-SIR2-mCherry-$T_{ENO2}$-LEU2, sir2::CgHIS3 | This study | NH1789 | See 'Strain and plasmid construction' for details |

| Reagent type (species) or resource | Designation | Source or reference | Identifiers | Additional information |
|---|---|---|---|---|
| Strain, strain background (S. cerevisiae) | yTRAP-NOP13, P$_{RPL18B}$-rtTA3-T$_{ADH1}$-TetO7-P$_{LEU2m}$-SIR2-mCherry-T$_{ENO2}$-LEU2, sir2::CgHIS3 | This study | NH1790 | See 'Strain and plasmid construction' for details |
| Strain, strain background (S. cerevisiae) | yTRAP-MRD1, P$_{RPL18B}$-rtTA3-T$_{ADH1}$-TetO7-P$_{LEU2m}$-SIR2-mCherry-T$_{ENO2}$-LEU2, sir2::CgHIS3 | This study | NH1804 | See 'Strain and plasmid construction' for details |
| Strain, strain background (S. cerevisiae) | yTRAP-RLP7, sir2::CgHIS3 | This study | NH1763 | See 'Strain and plasmid construction' for details |
| Strain, strain background (S. cerevisiae) | BY4741 MATa his3Δ1 leu2Δ0 met15Δ0 ura3Δ0, NHP6a-iRFP-kanMX, NOP15::P$_{NOP15}$-NOP15-mNeon-T$_{ADH1}$-URA3, SIS1-mCherry-HIS3 | This study | NH1752 | See 'Strain and plasmid construction' for details |
| Strain, strain background (S. cerevisiae) | BY4741 MATa his3Δ1 leu2Δ0 met15Δ0 ura3Δ0, NHP6a-iRFP-kanMX, NOP15::P$_{NOP15}$-NOP15-mNeon-T$_{ADH1}$-URA3, HSP104-mCherry-HIS | This study | NH1751 | See 'Strain and plasmid construction' for details |
| Strain, strain background (S. cerevisiae) | TMY3 MATa leu2-3,112 trp1-1 can1-100 ura3-1 ade2-1::LacI-GFP his3-11 rDNA::pTM-lacO50-URA3 | *Miyazaki and Kobayashi, 2011* | TMY3 | |

## Strain and plasmid construction

Standard methods for growth, maintenance, and transformation of yeast and bacteria were used throughout. The *S. cerevisiae* yeast strains used in this study were generated from the BY4741 strain background (MATa *his3Δ1 leu20Δ met15Δ0 ura3Δ0*). Yeast integrative transformations were performed using the standard lithium acetate method and confirmed by PCR. To make the Hsp104-GFP reporter, yEGFP-*HIS3* was amplified by PCR and integrated at the C-terminus of *HSP104* at the native locus by homologous recombination. To make the Sis1-mNeon reporter, pKT209 (ref) was subcloned to replace yEGFP with mNeon, and then mNeon-*URA3* was PCR-amplified and integrated into the C-terminus of *SIS1* at the native locus by homologous recombination. The plasmid pRS316-ΔssCPY*-GFP was generated in a previous study (*Park et al., 2007*). We subcloned into pRS306 by digesting both pRS306 vector and ΔssCPY*-GFP using SalI and HindIII and then ligating together. The newly assembled plasmid pRS306-ΔssCPY*-GFP was linearized using StuI and integrated into the *ura3-1* locus by homologous recombination. This strain background was BY4741 in which *ura3Δ0* was replaced by the W303 *ura3-1* locus (*Li et al., 2020*). The strain library used in the screen for RNA-binding protein aggregation was generated in a previous study (*Newby et al., 2017*).

To create the strains carrying each of the mNeon-tagged rRNA-binding proteins, we used Gibson assembly or traditional cloning to assemble plasmids containing the coding sequences for rRNA-binding proteins tagged C-terminally with mNeon under their native promoters. These plasmids were linearized using a restriction enzyme that would specifically cut within the promoter sequence, and then integrated into the genome by homologous recombination. Integration was verified using PCR and microscopy, and the copy number of the plasmid integrated was verified using PCR to ensure single-copy insertion. These strains were used to generate the data in *Figure 3A*. The rRNA-binding protein overexpression strains were similarly created. The plasmids containing the coding sequences for rRNA-binding proteins under their native promoters were constructed without any fluorescent protein tagging (to avoid potential interference of protein function by tags) and were also digested in the promoter region and integrated into the native genomic loci using homologous recombination. Integration and copy number were verified by PCR to ensure single-copy plasmid integration. These strains were used to generate the data in *Figure 3B*.

The *sir2Δ* mutant was created by amplifying either *CgURA3* or *CgHIS3* fragment to replace the *SIR2* open-reading frame by homologous recombination. Other genetic deletions were similarly created where each open-reading frame was replaced with a specific nutrient marker: *fob1Δ* was created using *CgURA3* or *CgHIS3*, *rpn4Δ* was created using *CgHIS3*, and *ubr2Δ* was created using *CgHIS3* or *CgLEU2*. The *HAP4* overexpression strain was constructed and verified previously (*Li et al., 2020*).

Because constitutive overexpression of *RRN3* or *NOP15* adversely affects cellular physiology and condition, we constructed doxycycline-inducible expression plasmids for both genes to conditionally

activate their overexpression during aging. For plasmid construction, we used the MoClo toolkit of yeast gene parts and followed the gene assembly strategy as described previously (*Lee et al., 2015*). Any Tet-On expression system consists of the same basic components: a constitutive promoter driving the transactivator rtTA expression and a promoter with a TetO element driving a gene of interest. More specifically, the ORF of *RRN3* or *NOP15* was PCR-amplified and assembled into the entry vector pYTK001 by Golden Gate Assembly. Then it was assembled with the following plasmids by Golden Gate Assembly to create a specific combination of promoter-ORF-fluorescent tag-terminator: pYTK003 (ConL1), NHB1016 (TetO7-$P_{LEU2m}$), pYTK-ORF (*RRN3* or *NOP15*), pYTK034 (mRuby2), pYTK065 ($T_{ENO2}$), pYTK072 (ConRE), and pYTK095 (AmpR-ColE1). Meanwhile, the coding sequence of rtTA3 was put under the $P_{RPL18B}$ promoter from the MoClo yeast toolkit, creating the $P_{RPL18B}$-*rtTA3* plasmid (NHB1028). The resulting two plasmids and NHB0980, which is a backbone plasmid containing *LEU2* as a nutritional selection marker, were then assembled by Golden Gate Assembly to get the final plasmid, $P_{RPL18B}$-*rtTA3*-$T_{ADH1}$-TetO7-$P_{LEU2m}$-*RRN3*-mRuby2-$T_{ENO2}$-*LEU2* (NHB1148) or $P_{RPL18B}$-*rtTA3*-$T_{ADH1}$-TetO7-$P_{LEU2m}$-*NOP15*-$T_{ENO2}$-*LEU2* (NHB1150). All the pYTK plasmids were from *Lee et al., 2015*. The inducible overexpression yeast strains were then created through genomic integration of NHB1148 or NHB1150 (linearized using *Notl*) by homologous recombination with the flanking sequences of the *leu2Δ0* locus.

## Microfluidic device fabrication

Design and fabrication of the microfluidic device for yeast replicative aging followed previously published work (*Jin et al., 2019*; *Li et al., 2020*; *Li et al., 2017*). In brief, 4-inch silicon wafers (University Wafer Inc) were patterned with SU8 2000 series (Kayakli Advanced Materials, Inc) photoresists using standard photolithography techniques in general accordance with the guidelines provided by the manufacturer. A polydimethylsiloxane (PDMS) device was made from the silicon wafer mold by mixing 33 g of Sylgard 184 (Dow Inc) and pouring it on the wafer surrounded with aluminum foil. The wafer and PDMS are then degassed in a vacuum chamber and cured on a level surface for at least 1 hr. Design and fabrication of the DynOMICS device were carried out using techniques described previously (*Graham et al., 2020*). This PDMS device was made from the silicon wafer mold by mixing 77 g of Sylgard 184 and pouring it on the wafer centered on a level 5″ × 5″ glass plate surrounded by an aluminum foil seal. The degassed PDMS is placed on a level surface and allowed to cure at 95°C for 1 hr.

## Single-cell aging microfluidics setup

A PDMS aging device was cleaned and sonicated in 100% ethanol for 15 min, followed by a rinse sonication in Milli-Q water. The device was dried and cleaned with an adhesive tape. A glass coverslip was cleaned in a series of washes with heptane, followed by methanol, then Milli-Q water, and subsequently dried with an air gun. Both the coverslip and the PDMS device were exposed to oxygen plasma to bond and create a fully assembled device. Once assembled, each single-cell aging device was inspected to ensure no defects or dust contamination were present.

To begin the experiment setup, the device was first placed under vacuum for 20 min. All media ports covered were then immediately covered by 0.075% Tween 20 for approximately 10 min. The device was then placed on an inverted microscope with a 30°C incubator system. Media ports were connected to plastic tubing and 60 mL syringes with fresh SCD media (prepared from CSM powder from Sunrise Science, #1001-100, with 2% glucose) medium containing 0.04% Tween-20. Initially the height of the syringes was approximately 2 ft above the microscope stage. The waste ports of the device were also connected to plastic tubing, which were attached by tape to stage height. Yeast cells were inoculated into 1.5 mL of SCD and cultured overnight at 30°C. This saturated overnight culture was then diluted 1:10,000 and grown at 30°C overnight until cells reached approximately OD600nm 0.6. For loading, cells were diluted approximately twofold and transferred to a 60 mL syringe (Luer-Lok Tip, BD) and connected to plastic tubing (TYGON, ID 0.020 IN, OD 0.060 IN, wall 0.020 IN). Cells were loaded by temporarily replacing the input media port with the syringe filled with yeast culture. The syringe containing the yeast is also placed approximately 2 ft above the stage. The media and cells flow into the device using gravity-driven flow. Cell traps are generally filled with cells in under a minute, at which point the loading tubing is replaced with the media tubing and syringe. Once cells are loaded, syringes are raised to be approximately 60 inches above the stage. Waste

tubing is lowered to the floor and waste is collected in a 50 mL tube to measure flow rate of about 2.5 mL/day. Note that Tween-20 is a non-ionic surfactant that helps reduce cell friction on the PDMS (*Ferry et al., 2011*). We have validated previously that this low concentration of Tween-20 has no significant effect on cellular lifespan or physiology. This setup protocol was developed and described in our previous studies (*Jin et al., 2019*; *Li et al., 2020*; *Li et al., 2017*).

## Time-lapse microscopy

Time-lapse microscopy experiments were performed using a Nikon Eclipse Ti-2 inverted fluorescence microscope with Perfect Focus and a back-illuminated sCMOS camera (Teledyne Photometrics Prime 95B). The light source is a Lumencor SpectraX. Images were taken using a CFI plan Apochromat Lambda DM ×60 oil immersion objective (NA 1.40 WD 0.13MM). Microfluidic devices were taped to a custom-built stage adapter and placed on the motorized stage. In all experiments, images were acquired using Nikon Elements software every 15 min for the duration of the yeast lifespan, typically 80 hr or longer, unless otherwise stated. The exposure and intensity settings were as follows for each of the fluorescence channels: GFP 10 ms with 10% light intensity, mCherry 50 ms with 5% light intensity, and Cy5 (iRFP) 200 ms with 2% light intensity.

## Confocal microscopy

Confocal images were acquired using a CSU-X1 spinning disk confocal module on the Nikon Eclipse Ti2-E scope used for aging experiments. The excitation light is controlled using an Agilent laser box with 405 nm, 488 nm, 561 nm, or 640 nm lasers. Laser light is focused through the microlenses of the spinning excitation disk (Yokogawa CSU-X1). Images were taken using either Plan Apo lambda ×60 NA 1.40 oil, or SR HP APO TIRF ×100 1.49 NA objectives. Laser intensity settings were 30% for 488 nm, 50% for 561 nm, and 50% for 640 nm. For time-lapse confocal imaging in aging cells, we acquire the images every 13 hr to minimize phototoxicity generated from confocal laser scanning.

## DynOMICS and yTRAP screen setup

Four 48-strain yeast DynOMICS devices were cleaned with 70% ethanol, DI water, and Scotch tape (3M), and each was aligned to a custom Singer ROTOR-compatible fixture. Both the fixture and a clean glass slide sonicated with 2% Hellmanex III were exposed to oxygen plasma. Cells were spotted from the previously arrayed agar plate to the aligned PDMS device using the Singer ROTOR spotting robot. The device and glass slide were bonded together and cured for 2 hr. Bonded chips were placed in a vacuum for 20 min before removal and covering of the inlet and outlet ports with SCD without folic acid or riboflavin media with 0.04% Tween-20. The microfluidic devices were then placed on two Nikon Ti microscopes, with two chips being placed on each Ti microscope. Media ports were then connected using plastic tubing and 60 mL syringes containing SCD media (without folic acid or riboflavin) with 0.04% Tween-20. At this point, we began imaging using a ×4 objective on each microscope as yeast cells began to grow to fill the 'bulb' regions of each position of the device and eventually fill the downstream 'HD biopixels.' Both scopes were equipped with CoolSnap HQ2 cameras (Photometrics). Initial imaging conditions on the first Ti scope containing half of the yTRAP library were as follows: phase contrast 10 ms, GFP 400 ms, and RFP 300 ms. For the second Ti microscope, phase contrast was 20 ms, GFP 600 ms, and RFP 300 ms. Approximately 2 days after initial setup of the experiment, yeast had grown to fill the downstream biopixels at each position across the four devices. At this point, the GFP exposure time for each scope was increased to 700 ms, and we marked this point as the starting point for future data analysis of fluorescent trajectories. Images were acquired every 20 min throughout the length of the experiment.

## Classification of Mode 1 and Mode 2 aging cells

We classified Mode 1 and Mode 2 aging cells based on the age-dependent changes in their daughter morphologies, as described in our previous study (*Li et al., 2020*). Mode 1 cells produce elongated daughters at the late stage of lifespan, whereas Mode 2 cells produce small rounded daughters until death. The classification was further confirmed by iRFP fluorescence (indicating the intracellular heme level) and cell cycle lengths during aging. Mode 1 and Mode 2 cells exhibit distinct dynamics of iRFP fluorescence during aging – the iRFP fluorescence increases toward the late stage of Mode 1 aging; in contrast, iRFP signal sharply decreases at the early stage of Mode 2 aging and remains extremely low

throughout the entire lifespan. Mode 1 and Mode 2 cells show age-dependent extension of cell cycle length with different timing and extents. Mode 1 cells show a gradual extension of cell cycle length at the late stages of aging; in contrast, Mode 2 cells show a much earlier and more dramatic extension of cell cycle length during aging (*Figure 1—figure supplement 1*; *Li et al., 2020*). These age-dependent iRFP fluorescence and cell cycle length dynamics provide robust and quantitative metrics to further confirm the classification of the two aging modes, independent of the need for specific microfluidic devices or imaging setup.

## Experiments with doxycycline-induced protein overexpression

Doxycycline was used as an activator to induce expression of TetO promoter-driven constructs including *RRN3* and *NOP15*. It was introduced into the media syringe to a final concentration of 2 μM. Doxycycline was delivered immediately after the cells were loaded into the microfluidic device and image acquisition began. Aging cells were exposed to doxycycline throughout the entire aging experiments. To control for the effect of doxycycline itself on Nop15 aggregation, strains lacking the inducible constructs (e.g., *fob1Δ*) were tested in parallel with and without doxycycline.

## Quantification and statistical analysis

Sample size for each experimental result can be found in the corresponding figure legends. To evaluate the statistical significance of lifespan differences, p-values were calculated using the Gehan–Breslow–Wilcoxon method, as performed in previous publications (*Crane et al., 2019*; *Li et al., 2020*), and are included in the corresponding RLS plots or the figure legends.

## Acknowledgements

We thank Dr. Ahmad S Khalil (Boston University) for generously providing us the yTRAP RBP sensor library, Dr. Dieter H Wolf (University of Stuttgart, Germany) for generously providing us the pRS316-Δss-CPY*-GFP plasmid, and Dr. Takehiko Kobayashi (University of Tokyo, Japan) for generously providing us the rDNA:lacO, LacI-GFP strain. This work was supported by National Institutes of Health R01 AG056440 (to NH, JH, LP, LST), GM111458 and AG068112 (to NH), NIH T32GM007240 (JP), and NSF MCB1716841 (LP).

## Additional information

### Funding

| Funder | Grant reference number | Author |
|---|---|---|
| National Institutes of Health | R01AG056440 | Lev S Tsimring Lorraine Pillus Jeff Hasty Nan Hao |
| National Institutes of Health | R01GM111458 | Nan Hao |
| National Institutes of Health | R01AG068112 | Nan Hao |
| National Institutes of Health | T32GM007240 | Julie Paxman |
| National Science Foundation | MCB1716841 | Lorraine Pillus |

The funders had no role in study design, data collection and interpretation, or the decision to submit the work for publication.

### Author contributions

Julie Paxman, Zhen Zhou, Conceptualization, Formal analysis, Investigation, Methodology, Writing - original draft, Writing - review and editing; Richard O'Laughlin, Formal analysis, Investigation,

Methodology, Writing - review and editing; Yuting Liu, Yang Li, Wanying Tian, Hetian Su, Yanfei Jiang, Shayna E Holness, Elizabeth Stasiowski, Formal analysis, Investigation, Writing - review and editing; Lev S Tsimring, Lorraine Pillus, Jeff Hasty, Conceptualization, Resources, Funding acquisition, Methodology, Writing - review and editing; Nan Hao, Conceptualization, Resources, Supervision, Funding acquisition, Methodology, Writing - original draft, Project administration, Writing - review and editing

### Author ORCIDs
Zhen Zhou ⬤ http://orcid.org/0000-0003-0706-9089
Yang Li ⬤ http://orcid.org/0000-0001-5714-0416
Shayna E Holness ⬤ http://orcid.org/0000-0001-6730-0583
Lev S Tsimring ⬤ http://orcid.org/0000-0003-0709-3548
Lorraine Pillus ⬤ http://orcid.org/0000-0002-8818-5227
Nan Hao ⬤ http://orcid.org/0000-0003-2857-4789

### Decision letter and Author response
Decision letter https://doi.org/10.7554/eLife.75978.sa1
Author response https://doi.org/10.7554/eLife.75978.sa2

## Additional files

### Supplementary files
• Transparent reporting form

### Data availability
All data generated or analysed during this study are included in the manuscript and supporting file. Source data have been provided for Figure 2.

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
