## [Editor Report]

Investigating the link between rDNA silencing and protein homeostasis, this study addresses an interesting and exciting question. The authors show how age-dependent loss of rDNA silencing contributes to protein aggregation. Importantly, the article furthers the understanding of distinct aging trajectories and raises important questions about how these processes might be relevant in multicellular organisms.

---

## [Decision Letter]

**Decision letter after peer review:**

Thank you for submitting your article "Age-dependent aggregation of ribosomal RNA-binding proteins links deterioration in chromatin stability with loss of proteostasis" for consideration by *eLife*. Your article has been reviewed by 3 peer reviewers, including Martin Sebastian Denzel as Reviewing Editor and Reviewer #1, and the evaluation has been overseen by Carlos Isales as the Senior Editor. The following individual involved in review of your submission has agreed to reveal their identity: Peter Tessarz (Reviewer #3).

Essential revisions:

The link between rDNA instability and protein aggregation is intriguing and important. The reviewers agree, however, that this link between two hallmarks of aging (which is the key selling point for the manuscript), is not fully supported and other explanations of the data are possible.

1. A clear and convincing causal link between rDNA instability and protein aggregation is needed.

2. It would be important to address if hsp104 puncta in fact indicate protein homeostasis defects.

3. The nature of the RBP aggregates remains unclear. Might they be deteriorating nucleoli?

*Reviewer #1 (Recommendations for the authors):*

To better support the conclusions made in the paper I would suggest the following experiments.

1. Loss of sir2 exacerbates hsp104 aggregation (Figure 1). To demonstrate a direct involvement, it would be great to also analyze sir2 o/e yeast. They have an extended lifespan and it would be interesting to know if they show reduced hsp104 aggregation.

2. NAM was used in the screen to inhibit sir2. It would be helpful to confirm some of the screen results in the sir2 mutant background. This is done for Nop15, but it would be good to confirm the results more generally, e.g. for the top 15 decreased RBPs.

3. Figure 4 analyses the contribution of rDNA circles and of rDNA transcription to Nop15 aggregation. These data are very important to illuminate the mechanistic link between rDNA silencing and protein aggregation. For this reason, it would be important to directly measure the levels of rRNA and of the rDNA circles.*Reviewer #2 (Recommendations for the authors):*

1. Previous studies showed that the formation of Hsp104 foci does not mean a proteostasis decline, instead, it is probably an adaptation or resolution to the proteostasis stress. For example, Barral's lab published an *eLife* paper in 2015 using microfluidics and imaging to show that protein aggregates associated with replicative aging without compromising protein quality control. In addition, the field generally considers protein aggregation as a protective solution that resolves proteostasis defect by concentrating toxic misfolded proteins into a deposition. Fail to form foci is correlated with cytotoxicity and death (e.g., Finkbeiner's 2004 nature paper, PMID15483602). This probably explains why the Model 1 cells, which show Hsp104 foci, actually have longer lifespan than the Model 2 cells that failed to assemble a Hsp104 foci.

2. It is not clear if the condensation of RBPs they showed are really protein aggregates. The example images in Figure 3A shows that these RBPs form several condensates instead of one that found in young and Model 2 cells. It is known that nucleolus fragment during aging, therefore, the condensates they saw are likely just an outcome of nucleolus fragmentation. This will make sense given that model 1 cells have rDNA silence defects, the accumulation of ERCs and their transcription help nucleate multiple condensates. By definition, protein aggregates accumulate misfolded proteins that do not carry their original functions. If these RBP condensates contains ERCs and there is active transcription of rRNA going on inside these RBP condensates, then they are not protein aggregates. Based on the logic of this manuscript, the RBP form multiple condensates because of rRNA transcription, which nucleate the condensation of these RBPs. If so, it implies that these condensates are active rRNA transcription center and/or ribosome assembly cores, but not protein aggregations. The authors should study the biophysical property of the condensates to see if cells with different number of condensates indeed have different condensates, such as FRAP etc, and compare them to the real protein aggregates such as Hsp104 foci. The other assays, such as the ones used by Barral's 2015 *eLife* paper (PMC4635334) to check if different chaperones are enriched on these condensates. This would be recommended for the authors to classify "moderate" vs "severe" condensates in figure 4.

3. There is not enough evidence that RBP condensation contributes to the cytosolic proteostasis reported by Hsp104 foci in their results. As explained below in #7, the % cells with multiple RBP condensates (in Figure 3 and Figure 4) are less than cells with Hsp104 foci (in Figure 1), which starts to appear at different life stage in wild type cells. Instead of investigating the temporal correlation between these RBP condensates and Hsp104 foci via single cell trajectories in wild type cells, the authors focus on the correlations between the different level of rDNA silence defect in mutant strains with different probability of showing Hsp104 foci in figure 5. However, it is not surprising to see short lived cells develop Hsp104 foci earlier while long lived strains delay their appearance. This correlation applies to other mutants that do not have direct effect on rDNA silence and ERC accumulation and therefore cannot exclusively support a casual effect here in their model. In the end, the reviewer cannot see how or whether rDNA instability causes proteostasis defect, especially the Hsp104 foci are in the cytosol. In fact, the authors do not consider the opposite hypothesis that the loss of proteostasis leads to the RBP condensation as shown in the rpn4∆ cells (figure 4).

In addition to these major concerns, there are several important things the authors have to resolve:

4. Figure 1 is critical to show the enrichment of Hsp104 foci in Model 1 cells and author implies that rDNA silence defect correlates with the Hsp104 foci. The example cells shown in Figure 1A shows that rDNA-GFP increase happens at age of 30 but the appearance of Hsp104 foci happens at 19. It is not clear to the reviewer how can rDNA instability contributes to hsp104 foci if the instability happens >10 generations later. I think the authors should include the information of rDNA-GFP fluctuation in the quantification showing in Figure 1B as well to quantitively show such correlation between rDNA-GFP and Hsp104 foci. A correlation between them should be calculated as well.

5. Figure 1-supplement 2 should show the trajectories of foci like Figure 1B as well. Being quantitive is the strength of single cell microfluidics and the authors should include the individual trajectory when the microfluidics data contain such information.

6. the correlation shown in Figure 1-supplement 3 is a result of mixed WT and ∆sir2. The author should calculate the correlation for each strain independently. The inclusion of ∆sir2 data point seems to increase the correlation shown in this figure. In the other words, the RLS and age for the first foci appearance in WT is much weaker. That was my impression when looking at Figure 1B as I can see the cells with longer RLS is not free of Hsp104 foci. Instead, these long-lived cells are associated with more Hsp104 foci. This goes back to my first point that Hsp104 foci is an indication of cellular adaptation but not defect.

7. Figure 3A needs quantification to show % cells showing these phenotypes. In fact, related to this question, the authors showed the trajectory of individual cells in figure 4A and I think the % of cells showing multiple RBP condensates (they refer to as "severe aggregation") are less than 5% before age 21-25. However, about 40% cells started to show Hsp104 foci around 10-15 generations and more cells with Hsp104 foci in figure 1A. If these multiple RBP condensates indeed contribute to Hsp104 foci formation in the cytosol, this connection is 10 generations apart. For this reason, it is highly recommended to analyze the trajectory of RBP condensation and rDNA-GFP/Hsp104 foci formation together for individual cells from the microfluidics to show/calculate temporal correlation.

8. Figure 4B has a quantification error: the ∆ubr2 cells have 1 out of 5 cells showing blue lines (so called "moderate aggregation") between 4680-5460 min in the single cell trajectory, but the quantification showing that cells with 26+ have 50% with blue lines (so called "moderate aggregation").

9. in page 11, line 3, it says the Nop15-mNeon was visualized every 13 hours? I think that's not right.

10. The authors use Rrn3 OE to show that increased rRNA production correlates with appearance of multiple RBP condensates. However, Rrn3 itself is a low complex protein (between a.a. 250-306 etc) and therefore its OE could seed/promote the condensation of other proteins like other people shown in liquid phase separation studies.

11. Figure 5C shows a synergistic effect between ubr2∆ and HAP4OE. The authors showed in their previous paper that HAP4 OE change the Model 2 cells, how about ubr2∆? From the logic presented in the manuscript, ubr2∆ should increase the RLS of Model 1 cells but not Model 2 cells. This is not tested.

*Reviewer #3 (Recommendations for the authors):*

The Hsp104-GFP foci shown throughout the paper are convincing. Given that the authors focus on rRNA-binding proteins later in the manuscript, it would be good to understand if the majority of foci are cytoplasmic or nuclear. Cytoplasmic foci might be more difficult to reconcile with aggregating nucleolar proteins.

sir2 enhances Hsp104 foci, but does fob1 decrease them – so, is enhanced stability able to prevent protein aggregation? As sir2D will impact other heterochromatic regions in the genome (mating type locus, telomeres), the authors would need to provide more evidence that the observed loss in proteostasis is indeed mediated by an increase in rDNA instability. One simple way would be to use strains that have lower rDNA copy numbers, which leads to instability specifically at this locus (see Ida et al., Science 2010). How would Hsp104-GFP look like in these strains?

Specificity of NAM-mediated Sir2 inhibition: NAM inhibits all NAD-depdendent deacetylases in yeast, incl. Hst1-4, thus will induce not only rDNA desilencing, but many other chromatin and non-chromatin dependent processes. Thus, this assay (specifically over the long time, in which cells were monitored) does not link rDNA instability and protein aggregation convincingly. Degrading sir2 by e.g. auxin-mediated degradation an/or fob1 degradation in a sir2D would provide more specificity.

In Figure 3A, the authors use mNeon-tagged copies of identified (and potentially aggregating) RBPs and track this at 80% lifespan. Tagged RBPs form some form of clusters. However, one issue with this experiment is the fact that these tagged proteins can also be seen as nucleolar markers and it is not clear from the figure whether the fluorescence is just marking deteriorating nucleoli or really protein aggregations. This is particularly true as the Hsp104-GFP staining only showed a very small bright focus. A co-localisation would be necessary to bridge the early observation with this experiment.

A similar concern holds true for the genetic dissection of Nop15 aggregation (Figure 4). The results are in line with the hypothesis, but it is not clear from the figures whether the authors monitor the nucleolus or indeed aggregation. Again, Hsp104- colocalisation might help.

The rRNA overexpression in an Rrn3 OE should be experimentally verified.

In Figure 5A, the effect of Rrn3 and Nop15 overexpression on Hsp104 foci formation is very subtle. It would be also necessary to show exemplary microscopy images and colocalisations to understand if Rrn3/Nop15 would trigger aggregate formation.

[Editors’ note: further revisions were suggested prior to acceptance, as described below.]

Thank you for resubmitting your work entitled "Age-dependent aggregation of ribosomal RNA-binding proteins links deterioration in chromatin stability with challenges to proteostasis" for further consideration by *eLife*. Your revised article has been evaluated by Carlos Isales (Senior Editor) and a Reviewing Editor.

The manuscript has been substantially improved but there are some remaining issues that need to be addressed, as outlined below:

The reviewers agree that the current dataset cannot support a causal sequence between nucleolar expansion and the degeneration of proteostasis. Studying the temporal order of Nop15 condensation vs Hsp104 aggregation within the same cell is required to answer this correlation.

---

## [Author Response]

Essential revisions:The link between rDNA instability and protein aggregation is intriguing and important. The reviewers agree, however, that this link between two hallmarks of aging (which is the key selling point for the manuscript), is not fully supported and other explanations of the data are possible.1. A clear and convincing causal link between rDNA instability and protein aggregation is needed.

To address this concern, we included new experiments and consideration of the existing literature:

1. We monitored Nop15-mNeon and Hsp104-mCherry in the same cells (new Figure 5A). We found continuous co-occurrence of Nop15 aggregation and Hsp104 foci during the later stage of aging in the majority of Mode 1 cells (Figure 5A, left). Furthermore, in 66% of these cells, Nop15 aggregation (indicated by green bars in Figure 5A) immediately preceded the co-occurrence phase (indicated by yellow bars in Figure 5A), *suggesting* that rRNA-binding protein aggregation leads to global proteostasis stress in a large fraction of aging cells. These new data, together with the results showing that genetic perturbations of the rDNA stability pathway affect Hsp104 foci formation (Figure 5B), support a causal link between rDNA instability and protein aggregation in aging.

2. We performed experiments to examine whether the yTRAP screen results upon NAM treatment are mediated specifically through Sir2. For the top 5 rRNA-binding protein responders, we deleted the endogenous copy of *SIR2* and introduced a doxycycline-controlled promoter system for Sir2 expression in each of the yTRAP sensor strains. We found that the absence of Sir2 expression promoted aggregation of all 5 rRNA-binding proteins tested (new Figure 2–figure supplement 2), confirming the specificity of Sir2’s role.

3. We examined Hsp104 foci during aging in additional mutants that influence rDNA stability. Specifically, we monitored Hsp104 foci formation during the aging process in the *hap4*D strain, which is short-lived due to mitochondrial defect but has enhanced rDNA stability (Li et al., 2020). We observed strikingly decreased Hsp104 foci formation, compared to WT. Furthermore, deletion of Sir2 in the *hap4*D strain had a modest effect on the lifespan, but substantially *increased* Hsp104 foci formation (new Figure 1–figure supplement 4). These results exclude the possibility that *sir2*D exacerbates Hsp104 aggregation simply because it is short-lived and further confirm the role of Sir2 in protecting from proteostasis stress.

4. We monitored changes in rDNA copy number during aging using a rDNA::lacO, LacI-GFP strain from the Kobayashi group (Miyazaki and Kobayashi, 2011) and showed its temporal relationship with rRNA-binding protein aggregation (new Figure 4–figure supplement 1)..

5. We quantified the single-cell aging trajectories for aggregation of the nuclear chaperone Sis1 (new Figure 1C and D) and showed the colocalization of Sis1 in rRNA-binding protein aggregates (new Figure 3B). These results demonstrate the contribution of rRNA-binding protein aggregation to proteostasis stress.

6. We included single-cell aging trajectories for Hsp104 foci formation in *fob1*D, *fob1*D+*RNR3* o/e, *fob1*D+*NOP15* o/e, to better demonstrate the effects of Rnr3 and Nop15 overexpression on Hsp104 foci formation (new Figure 5–figure supplement 1).

7. We largely revised our working model (new Figure 6A) and the main text throughout the manuscript (see the point-to-point responses below for details).

2. It would be important to address if hsp104 puncta in fact indicate protein homeostasis defects.

We agree with the Reviewers that Hsp104 foci indicate the presence of “proteostasis stress” or “challenges to proteostasis” during aging, rather than “proteostasis decline” or “loss of proteostasis”. To address this:

1. We have revised the text and data interpretation throughout the manuscript (see tracked changes throughout the main text) and our working model (new Figure 6A) accordingly.

2. We quantified the single-cell aging trajectories for aggregation of the nuclear chaperone Sis1, as an additional proteostasis stress reporter in the nucleus (new Figure 1, C and D).

3. We replotted the correlation between first Hsp104 foci appearance and lifespan for WT and *sir2*∆ separately (new Figure 1–figure supplement 3).

3. The nature of the RBP aggregates remains unclear. Might they be deteriorating nucleoli?

We monitored the localization of Nop15, a representative rRNA-binding protein, and Sis1, the Hsp40 co-chaperone that functions in clearance of misfolded proteins in the nucleus. We showed that Sis1 clearly accumulated in Nop15 condensates in aged cells, whereas no such colocalization was observed in young cells (new Figure 3B). These results, together with the data showing (1) increased rRNA-binding protein aggregation shortens lifespan (Figre 3C), and (2) the proteasome capacity influences the kinetics and extent of rRNA-binding protein condensation (Fig. 4B), indicate that age-induced rRNA-binding protein condensates are *bona fide* protein aggregates.

Reviewer #1 (Recommendations for the authors):To better support the conclusions made in the paper I would suggest the following experiments.1. Loss of sir2 exacerbates hsp104 aggregation (Figure 1). To demonstrate a direct involvement, it would be great to also analyze sir2 o/e yeast. They have an extended lifespan and it would be interesting to know if they show reduced hsp104 aggregation.

We thank the reviewer for this suggestion. Along the same line as this suggestion, we monitored Hsp104 aggregation in *fob1*D (Figure 5B; Figure 5 —figure supplement 1A). *fob1*D enhances rDNA stability and extends the lifespan, similar to Sir2 o/e, while its effect is more specific to the rDNA stability pathway and ERC formation. We observed dramatically reduced Hsp104 foci formation in *fob1*D, which we feel is sufficient to demonstrate that enhancing rDNA stability reduces proteostasis stress.

2. NAM was used in the screen to inhibit sir2. It would be helpful to confirm some of the screen results in the sir2 mutant background. This is done for Nop15, but it would be good to confirm the results more generally, e.g. for the top 15 decreased RBPs.

As suggested by the reviewer, we performed experiments to examine whether the yTRAP screen results upon NAM treatment are mediated specifically through Sir2. For the top 5 rRNA-binding protein responders, we deleted the endogenous copy of *SIR2* and introduced a doxycyclinecontrolled promoter system for Sir2 expression in each of the yTRAP sensor strains. We found that the absence of Sir2 expression promoted aggregation of all 5 rRNA-binding proteins tested (new Figure 2 —figure supplement 2), confirming the specificity of Sir2’s role.

3. Figure 4 analyses the contribution of rDNA circles and of rDNA transcription to Nop15 aggregation. These data are very important to illuminate the mechanistic link between rDNA silencing and protein aggregation. For this reason, it would be important to directly measure the levels of rRNA and of the rDNA circles.

As suggested by the reviewer, we monitored changes in rDNA copy number (as an indication of the number of rDNA circles; see PMID:31291577) during aging using a rDNA::lacO, LacI-GFP strain from the Kobayashi group (Miyazaki and Kobayashi, 2011) and showed its temporal relationship with rRNA-binding protein aggregation (new Figure 4 —figure supplement 1). The same approach has been used to show age-induced increases in rDNA circles during yeast aging (see Morlot et al., 2019; PMID:31291577).

Reviewer #2 (Recommendations for the authors):1. Previous studies showed that the formation of Hsp104 foci does not mean a proteostasis decline, instead, it is probably an adaptation or resolution to the proteostasis stress. For example, Barral's lab published an eLife paper in 2015 using microfluidics and imaging to show that protein aggregates associated with replicative aging without compromising protein quality control. In addition, the field generally considers protein aggregation as a protective solution that resolves proteostasis defect by concentrating toxic misfolded proteins into a deposition. Fail to form foci is correlated with cytotoxicity and death (e.g., Finkbeiner's 2004 nature paper, PMID15483602). This probably explains why the Model 1 cells, which show Hsp104 foci, actually have longer lifespan than the Model 2 cells that failed to assemble a Hsp104 foci.

We agree with the reviewer that Hsp104 foci indicate the presence of “proteostasis stress” or “challenges to proteostasis” during aging, rather than “loss of proteostasis”. To address this, we have revised the text and data interpretation throughout the manuscript (see tracked changes throughout the main text) and our working model (new Figure 6A) accordingly.

2. It is not clear if the condensation of RBPs they showed are really protein aggregates. The example images in Figure 3A shows that these RBPs form several condensates instead of one that found in young and Model 2 cells. It is known that nucleolus fragment during aging, therefore, the condensates they saw are likely just an outcome of nucleolus fragmentation. This will make sense given that model 1 cells have rDNA silence defects, the accumulation of ERCs and their transcription help nucleate multiple condensates. By definition, protein aggregates accumulate misfolded proteins that do not carry their original functions. If these RBP condensates contains ERCs and there is active transcription of rRNA going on inside these RBP condensates, then they are not protein aggregates. Based on the logic of this manuscript, the RBP form multiple condensates because of rRNA transcription, which nucleate the condensation of these RBPs. If so, it implies that these condensates are active rRNA transcription center and/or ribosome assembly cores, but not protein aggregations. The authors should study the biophysical property of the condensates to see if cells with different number of condensates indeed have different condensates, such as FRAP etc, and compare them to the real protein aggregates such as Hsp104 foci. The other assays, such as the ones used by Barral's 2015 eLife paper (PMC4635334) to check if different chaperones are enriched on these condensates. This would be recommended for the authors to classify "moderate" vs "severe" condensates in figure 4.

We thank the reviewer for raising this question. As suggested, we monitored the localization of Nop15, a representative rRNA-binding protein, and Sis1, the Hsp40 co-chaperone that functions in clearance of misfolded proteins in the nucleus. We showed that Sis1 clearly accumulated in Nop15 condensates in aged cells, whereas no such colocalization was observed in young cells (new Figure 3B). These results, together with the data showing (1) increased rRNA-binding protein aggregation shortens lifespan (Figure 3C), and (2) the proteasome capacity influences the kinetics and extent of rRNA-binding protein condensation (Figure 4B), indicate that age-induced rRNAbinding protein condensates are *bona fide* protein aggregates.

3. There is not enough evidence that RBP condensation contributes to the cytosolic proteostasis reported by Hsp104 foci in their results. As explained below in #7, the % cells with multiple RBP condensates (in Figure 3 and Figure 4) are less than cells with Hsp104 foci (in Figure 1), which starts to appear at different life stage in wild type cells. Instead of investigating the temporal correlation between these RBP condensates and Hsp104 foci via single cell trajectories in wild type cells, the authors focus on the correlations between the different level of rDNA silence defect in mutant strains with different probability of showing Hsp104 foci in figure 5. However, it is not surprising to see short lived cells develop Hsp104 foci earlier while long lived strains delay their appearance. This correlation applies to other mutants that do not have direct effect on rDNA silence and ERC accumulation and therefore cannot exclusively support a casual effect here in their model. In the end, the reviewer cannot see how or whether rDNA instability causes proteostasis defect, especially the Hsp104 foci are in the cytosol. In fact, the authors do not consider the opposite hypothesis that the loss of proteostasis leads to the RBP condensation as shown in the rpn4∆ cells (figure 4).

As suggested by the reviewer, we monitored Nop15-mNeon and Hsp104-mCherry in the same cells (new Figure 5A). We found continuous co-occurrence of Nop15 aggregation and Hsp104 foci during the later stage of aging in the majority of Mode 1 cells (Figure 5A, left). Furthermore, in 66% of these cells, Nop15 aggregation (indicated by green bars in Figure 5A) immediately preceded the co-occurrence phase (indicated by yellow bars in Figure 5A), suggesting that rRNA-binding protein aggregation leads to global proteostasis stress in a large fraction of aging cells. We also observed occasional, transient appearance of Hsp104 foci during the early stage of aging in both Mode 1 and Mode 2 cells, which did not show any obvious relationship to Nop15 aggregation that occurred much later in aging. We have added a discussion about the potential sources of these early-life events (Page 17, Line 14-17).

To confirm that the different patterns of Hsp104 foci formation observed in various mutants (e.g. *sir2*D and *fob1*D) are not a side-effect of lifespan modulation, we examined Hsp104 foci during aging in additional mutants that influence rDNA stability. Specifically, we monitored Hsp104 foci formation during the aging process in the *hap4*D strain, which is short-lived due to mitochondrial defect but has enhanced rDNA stability (Li et al., 2020). We observed strikingly decreased Hsp104 foci formation, compared to WT. Furthermore, deletion of Sir2 in the *hap4*D strain had a modest effect on the lifespan, but substantially *increased* Hsp104 foci formation (new Figure 1 —figure supplement 4). These results exclude the possibility that *sir2*D exacerbates Hsp104 aggregation simply because it is short-lived and further confirm the role of Sir2 in protecting from proteostasis stress.

Furthermore, we have revised our working model (Figure 6A) and the main text (Page 16, Line 1-3) to discuss the potential regulation of rRNA-binding protein aggregation by the proteostasis network.

In addition to these major concerns, there are several important things the authors have to resolve:4. Figure 1 is critical to show the enrichment of Hsp104 foci in Model 1 cells and author implies that rDNA silence defect correlates with the Hsp104 foci. The example cells shown in Figure 1A shows that rDNA-GFP increase happens at age of 30 but the appearance of Hsp104 foci happens at 19. It is not clear to the reviewer how can rDNA instability contributes to hsp104 foci if the instability happens >10 generations later. I think the authors should include the information of rDNA-GFP fluctuation in the quantification showing in Figure 1B as well to quantitively show such correlation between rDNA-GFP and Hsp104 foci. A correlation between them should be calculated as well.

Hsp104-GFP and rDNA-GFP were not measured in the same cells or in the same experiments, and hence cannot be directly compared. We included representative images of rDNA-GFP in the previous version of Figure 1 to illustrate the difference in rDNA silencing between Mode 1 and Mode 2 aging cells. To avoid confusion, we removed the representative images of rDNA-GFP in new Figure 1.

5. Figure 1-supplement 2 should show the trajectories of foci like Figure 1B as well. Being quantitive is the strength of single cell microfluidics and the authors should include the individual trajectory when the microfluidics data contain such information.

As suggested by the reviewer, we quantified the single-cell aging trajectories for aggregation of the nuclear chaperone Sis1, as an additional proteostasis stress reporter in the nucleus (new Figure 1, C and D).

6. The correlation shown in Figure 1-supplement 3 is a result of mixed WT and ∆sir2. The author should calculate the correlation for each strain independently. The inclusion of ∆sir2 data point seems to increase the correlation shown in this figure. In the other words, the RLS and age for the first foci appearance in WT is much weaker. That was my impression when looking at Figure 1B as I can see the cells with longer RLS is not free of Hsp104 foci. Instead, these long-lived cells are associated with more Hsp104 foci. This goes back to my first point that Hsp104 foci is an indication of cellular adaptation but not defect.

As suggested by the reviewer, we replotted the correlation between first Hsp104 foci appearance and lifespan for WT and *sir2*∆ separately (new Figure 1 —figure supplement 3).

7. Figure 3A needs quantification to show % cells showing these phenotypes. In fact, related to this question, the authors showed the trajectory of individual cells in figure 4A and I think the % of cells showing multiple RBP condensates (they refer to as "severe aggregation") are less than 5% before age 21-25. However, about 40% cells started to show Hsp104 foci around 10-15 generations and more cells with Hsp104 foci in figure 1A. If these multiple RBP condensates indeed contribute to Hsp104 foci formation in the cytosol, this connection is 10 generations apart. For this reason, it is highly recommended to analyze the trajectory of RBP condensation and rDNA-GFP/Hsp104 foci formation together for individual cells from the microfluidics to show/calculate temporal correlation.

As addressed above in pt3, we monitored Nop15-mNeon and Hsp104-mCherry in the same cells (new Figure 5A) and observed that Nop15 aggregation (indicated by green bars in Figure 5A) immediately preceded continuous co-appearance of Nop15 aggregation and Hsp104 aggregation during later stages of aging in a large fraction of cells.

8. Figure 4B has a quantification error: the ∆ubr2 cells have 1 out of 5 cells showing blue lines (so called "moderate aggregation") between 4680-5460 min in the single cell trajectory, but the quantification showing that cells with 26+ have 50% with blue lines (so called "moderate aggregation").

Thanks for the reviewer for pointing this out. We have confirmed the accuracy of quantification.

9. In page 11, line 3, it says the Nop15-mNeon was visualized every 13 hours? I think that's not right.

For time-lapse confocal imaging in aging cells, we acquire the images every 13 hours to minimize phototoxicity generated from confocal laser scanning. We have added a clarification in the Methods (Page 29, Line 16-18).

10. The authors use Rrn3 OE to show that increased rRNA production correlates with appearance of multiple RBP condensates. However, Rrn3 itself is a low complex protein (between a.a. 250-306 etc) and therefore its OE could seed/promote the condensation of other proteins like other people shown in liquid phase separation studies.

Proteins with putative low complexity domains do not necessarily affect proteostasis or aggregation of other proteins, and we cannot find any previous evidence in support of the possibility that Rrn3 forms aggregates or contributes to proteostasis decline. Therefore, we stand with our interpretation of the data and our working model, which are supported by a series of other data together.

11. Figure 5C shows a synergistic effect between ubr2∆ and HAP4OE. The authors showed in their previous paper that HAP4 OE change the Model 2 cells, how about ubr2∆? From the logic presented in the manuscript, ubr2∆ should increase the RLS of Model 1 cells but not Model 2 cells. This is not tested.

We have added a discussion about the effects of *UBR2* deletion in lifespans in Mode 1 and Mode 2 cells and proteostasis in cells aging with mitochondrial defects in the text (Page 17, Line 22 – Page 18, Line 2).

Reviewer #3 (Recommendations for the authors):The Hsp104-GFP foci shown throughout the paper are convincing. Given that the authors focus on rRNA-binding proteins later in the manuscript, it would be good to understand if the majority of foci are cytoplasmic or nuclear. Cytoplasmic foci might be more difficult to reconcile with aggregating nucleolar proteins.

We thank the reviewer for raising this issue. We indeed observed that, similar to Hsp104, the nuclear chaperone Sis1 also forms aggregates predominantly in Mode 1 aging cells (new Figure 1, C and D). In addition, Sis1 aggregates colocalize with Nop15 aggregates in aged cells (new Figure 3B). Our working model to interpret the Hsp104 data is that rRNA-binding protein aggregation contributes to global proteostasis stress by exacerbating ribosomal dysfunction and increasing the burden on the proteostasis network, including chaperons and proteasomes (Page 15, Line 16 – Page 16, Line 1), as demonstrated in many examples from previous studies (Andersson et al. 2013, PMID: 24243762; Bence et al. 2001, PMID: 11375494; Outeiro and Lindquist, 2003, PMID: 14657500; Stefani and Dobson, 2003, PMID: 12942175; Verhoef et al. 2002, PMID: 12374759; Yu et al. 2019, PMID: 30936201).

sir2 enhances Hsp104 foci, but does fob1 decrease them – so, is enhanced stability able to prevent protein aggregation? As sir2D will impact other heterochromatic regions in the genome (mating type locus, telomeres), the authors would need to provide more evidence that the observed loss in proteostasis is indeed mediated by an increase in rDNA instability. One simple way would be to use strains that have lower rDNA copy numbers, which leads to instability specifically at this locus (see Ida et al., Science 2010). How would Hsp104-GFP look like in these strains?

We thank the reviewer for raising this question. We monitored Hsp104 aggregation in *fob1*D and observed dramatically reduced Hsp104 foci formation (Figure 5B; Figure 5 —figure supplement 1A). Since the effect of *fob1*D is more specific to the rDNA stability pathway and ERC formation than *sir2*D, we feel that this data is sufficient to demonstrate that enhancing rDNA stability reduces proteostasis stress.

Specificity of NAM-mediated Sir2 inhibition: NAM inhibits all NAD-depdendent deacetylases in yeast, incl. Hst1-4, thus will induce not only rDNA desilencing, but many other chromatin and non-chromatin dependent processes. Thus, this assay (specifically over the long time, in which cells were monitored) does not link rDNA instability and protein aggregation convincingly. Degrading sir2 by e.g. auxin-mediated degradation an/or fob1 degradation in a sir2D would provide more specificity.

As suggested by the reviewer, we performed experiments to examine whether the yTRAP screen results upon NAM treatment are mediated specifically through Sir2. For the top 5 rRNA-binding protein responders, we deleted the endogenous copy of *SIR2* and introduced a doxycyclinecontrolled promoter system for Sir2 expression in each of the yTRAP sensor strains. We found that the absence of Sir2 expression promoted aggregation of all 5 rRNA-binding proteins tested (new Figure 2 —figure supplement 2), confirming the specificity of Sir2’s role.

In Figure 3A, the authors use mNeon-tagged copies of identified (and potentially aggregating) RBPs and track this at 80% lifespan. Tagged RBPs form some form of clusters. However, one issue with this experiment is the fact that these tagged proteins can also be seen as nucleolar markers and it is not clear from the figure whether the fluorescence is just marking deteriorating nucleoli or really protein aggregations. This is particularly true as the Hsp104-GFP staining only showed a very small bright focus. A co-localisation would be necessary to bridge the early observation with this experiment.

We thank the reviewer for this suggestion. We monitored the localization of Nop15, a representative rRNA-binding protein, and Sis1, the Hsp40 co-chaperone that functions in clearance of misfolded proteins in the nucleus. We showed that Sis1 clearly accumulated in Nop15 condensates in aged cells, whereas no such colocalization was observed in young cells (new Figure 3B). These results, together with the data showing (1) increased rRNA-binding protein aggregation shortens lifespan (Figure 3C), and (2) the proteasome capacity influences the kinetics and extent of rRNA-binding protein condensation (Figure 4B), indicate that age-induced rRNAbinding protein condensates are *bona fide* protein aggregates.

A similar concern holds true for the genetic dissection of Nop15 aggregation (Figure 4). The results are in line with the hypothesis, but it is not clear from the figures whether the authors monitor the nucleolus or indeed aggregation. Again, Hsp104- colocalisation might help.

As addressed above, we showed colocalization of Nop15 aggregation and Sis1 aggregation in aged cells (new Figure 3B), indicating that age-induced rRNA-binding protein condensates are *bona fide* protein aggregates.

The rRNA overexpression in an Rrn3 OE should be experimentally verified.

We thank the reviewer for raising this up. It has been experimentally demonstrated that Rrn3 OE with a similar inducible system as we used can dramatically enhance rRNA synthesis (Figure 5C in PMID: 20321203). We have cited this paper in the text.

In Figure 5A, the effect of Rrn3 and Nop15 overexpression on Hsp104 foci formation is very subtle. It would be also necessary to show exemplary microscopy images and colocalisations to understand if Rrn3/Nop15 would trigger aggregate formation.

We note that we observed very dramatic effects of Rrn3 or Nop15 overexpression on Hsp104 foci formation in the *fob1*D background where ERC is abolished (Figure 5B, compare *fob1∆* + *RRN3* o/e and *fob1∆* + *NOP15* o/e with *fob1∆* alone). We included now single-cell aging trajectories for Hsp104 foci formation in *fob1*D, *fob1*D+*RNR3* o/e, *fob1*D+*NOP15* o/e, to better demonstrate the effects of Rnr3 and Nop15 overexpression on Hsp104 foci formation (new Figure 5 —figure supplement 1).

[Editors’ note: further revisions were suggested prior to acceptance, as described below.]

The reviewers agree that the current dataset cannot support a causal sequence between nucleolar expansion and the degeneration of proteostasis. Studying the temporal order of Nop15 condensation vs Hsp104 aggregation within the same cell is required to answer this correlation.

As suggested by the reviewers, we monitored Nop15-mNeon and Hsp104-mCherry in the same cells (new Figure 5A). We found continuous co-occurrence of Nop15 aggregation and Hsp104 foci during the later stage of aging in the majority of Mode 1 cells (Figure 5A, left). Furthermore, in 66% of these cells, Nop15 aggregation (indicated by green bars in Figure 5A) immediately preceded the co-occurrence phase (indicated by yellow bars in Figure 5A), *suggesting* that rRNA-binding protein aggregation leads to global proteostasis stress in a large fraction of aging cells.

We also observed occasional, transient appearance of Hsp104 foci during the early stage of aging in both Mode 1 and Mode 2 cells, which did not show any obvious relationship to Nop15 aggregation that occurred much later in aging. We have added a discussion about the potential sources of these early-life events (Page 17, Line 14-17).